# The *Imageable Genome*

**Pablo Jané** [1,2,8], **Xiaoying Xu** [3,8], **Vincent Taelman**[3,8], **Eduardo Jané** [4], **Karim Gariani**[5], **Rebecca A. Dumont**[3], **Yonathan Garama**[3], **Francisco Kim** [3], **María del Val Gomez**[6] & **Martin A. Walter** [3,7] ✉

Understanding human disease on a molecular level, and translating this understanding into targeted diagnostics and therapies are central tenets of molecular medicine[1]. Realizing this doctrine requires an efficient adaptation of molecular discoveries into the clinic. We present an approach to facilitate this process by describing the *Imageable Genome*, the part of the human genome whose expression can be assessed via molecular imaging. Using a deep learning-based hybrid human-AI pipeline, we bridge individual genes and their relevance in human diseases with specific molecular imaging methods. Cross-referencing the *Imageable Genome* with RNA-seq data from over 60,000 individuals reveals diagnostic, prognostic and predictive imageable genes for a wide variety of major human diseases. Having both the critical size and focus to be altered in its expression during the development and progression of any human disease, the *Imageable Genome* will generate new imaging tools that improve the understanding, diagnosis and management of human diseases.

Molecular imaging has the unique ability to non-invasively, quantitatively and reproducibly assess target molecule expression on cellular and subcellular level[2]. It is an integral element of molecular medicine that is designed to deepen the understanding of human health and disease[3]. One of the most widely utilized molecular imaging modalities in the clinic is positron emission tomography (PET), which shares molecular targets with other widespread modalities such as magnetic resonance imaging. PET's ability to provide a total-body readout in serial exams without sampling error make it an ideal imaging tool, particularly for the domains of neurology, cardiology and oncology.

The combination of PET with computed tomography (CT) and the radiotracer fluorine-18-fluorodeoxyglucose (FDG) established the role of PET-CT as a clinical diagnostic workhorse, notably in oncology[4]. Yet, the gateway to unlocking the full potential of PET in visualizing spatiotemporal pathobiology, characterizing resistance-generating genetic transformations and identifying individual disease progression lies in the accurate targeting of molecular markers, many of which can be assessed via the thousands of already existing radiotracers. These radiotracers, compiled in the NIH Molecular Imaging and Contrast Agent Database (MICAD), represent the ultimate key to develop PET into a clinically relevant diagnostic, predictive and prognostic tool for all types of human diseases, as well as a driving force behind the realization of personalized molecular medicine into routine clinical use.

Recently, PET-based molecular imaging has had spectacular successes with the development of several radiotracers that target specific tumour cells or cells of the tumour environment, such as PSMA or FAPI[5,6]. However, despite decades and billions of dollars spent on research[7], still less than 1% of developed radiotracers arrive in the clinic. One reason for this translational bottleneck is linked to the fundamental lack of knowledge concerning the entirety of molecules that can be targeted with the repertoire of available molecular imaging agents, as well as the relevance of these molecules across the spectrum of human disease.

Here we attempt to close this gap by providing a global view on the field of molecular imaging, and by identifying the entirety of gene products that can be targeted with molecular imaging. In doing so, we describe the *Imageable Genome*, the part of the human genome whose expression can be assessed via molecular imaging. By correlating the *Imageable Genome* with recent genomic datasets, we subsequently

[1]University of Geneva, Geneva, Switzerland. [2]Nuclear Medicine and Molecular Imaging Division, Geneva University Hospitals, Geneva, Switzerland. [3]University of Lucerne, Lucerne, Switzerland. [4]Departamento de Matemática Aplicada a la Ingeniería Aeroespacial - ETSIAE, Universidad Politécnica de Madrid, 28040 Madrid, Spain. [5]Division of Endocrinology, Diabetes, Nutrition and Patient Therapeutic Education, Geneva University Hospitals, Geneva, Switzerland. [6]Servicio de Medicina Nuclear, Hospital Universitario Ramón y Cajal, Madrid, Spain. [7]St. Anna Hospital, University of Lucerne, Lucerne, Switzerland. [8]These authors contributed equally: Pablo Jané, Xiaoying Xu, Vincent Taelman. ✉e-mail: martin.alexander.walter@gmail.com

demonstrate its transformative potential for the fields of neurology, cardiology and oncology. In developing and employing this novel data pipeline, we transgress the borders between medical imaging, genomics, systems biology and data science to advance the understanding, diagnosis and treatment of human diseases.

## Results

### Data pipeline

Using machine learning and deep learning, we created a hybrid human-AI pipeline that connects the combined data in medicine (PubMed), molecular imaging (NIH MICAD), tissue gene expression (GEO) and gene–disease association (DisGeNET) with individual clinical data and gene expression profiles of 16,327 patients and 50,403 healthy controls (Fig. 1). We built a pipeline that is simple, robust, easy to deploy, and capable of dealing with large data sets, while optimizing its code readability, logical structure and the use of well-known languages, including Python and Structured Query Language (SQL).

We first downloaded and parsed the entire baseline MEDLINE/PubMed dataset consisting of 33.4 million entries and 22.5 million abstracts, as well as the NIH Molecular Imaging and Contrast Agent

Database (MICAD) comprising 4531 PubMed entries and 5360 molecular imaging agents. We then cross-referenced both datasets, and implemented and trained a convolutional neural network for natural language processing, i.e., a text classifier that identifies PubMed entries on molecular imaging with high accuracy (Fig. 1a). Subsequently, we implemented and trained a second convolutional neural network for natural language processing, i.e., a named entity recognizer that extracts radiotracers and their target protein from the PubMed entries on molecular imaging with high accuracy (Fig. 1b and Supplementary Data 1). We then developed filters that identify clinically used radiotracers and tracer-to-protein associations, thereby defining the *Imageable Proteome*. Finally, we translated the names of proteins to those of the coding genes, thereby establishing the *Imageable Genome*.

### The *Imageable Genome*

Our pipeline identified 6387 publications describing 9285 radiotracer-to-gene associations. These radiotracers target the products of 1173 imageable genes, which constitute the *Imageable Genome* (Supplementary Data 2). The *Imageable Genome* is constantly growing, and currently comprises 1166 genes located on all chromosomes but the Y

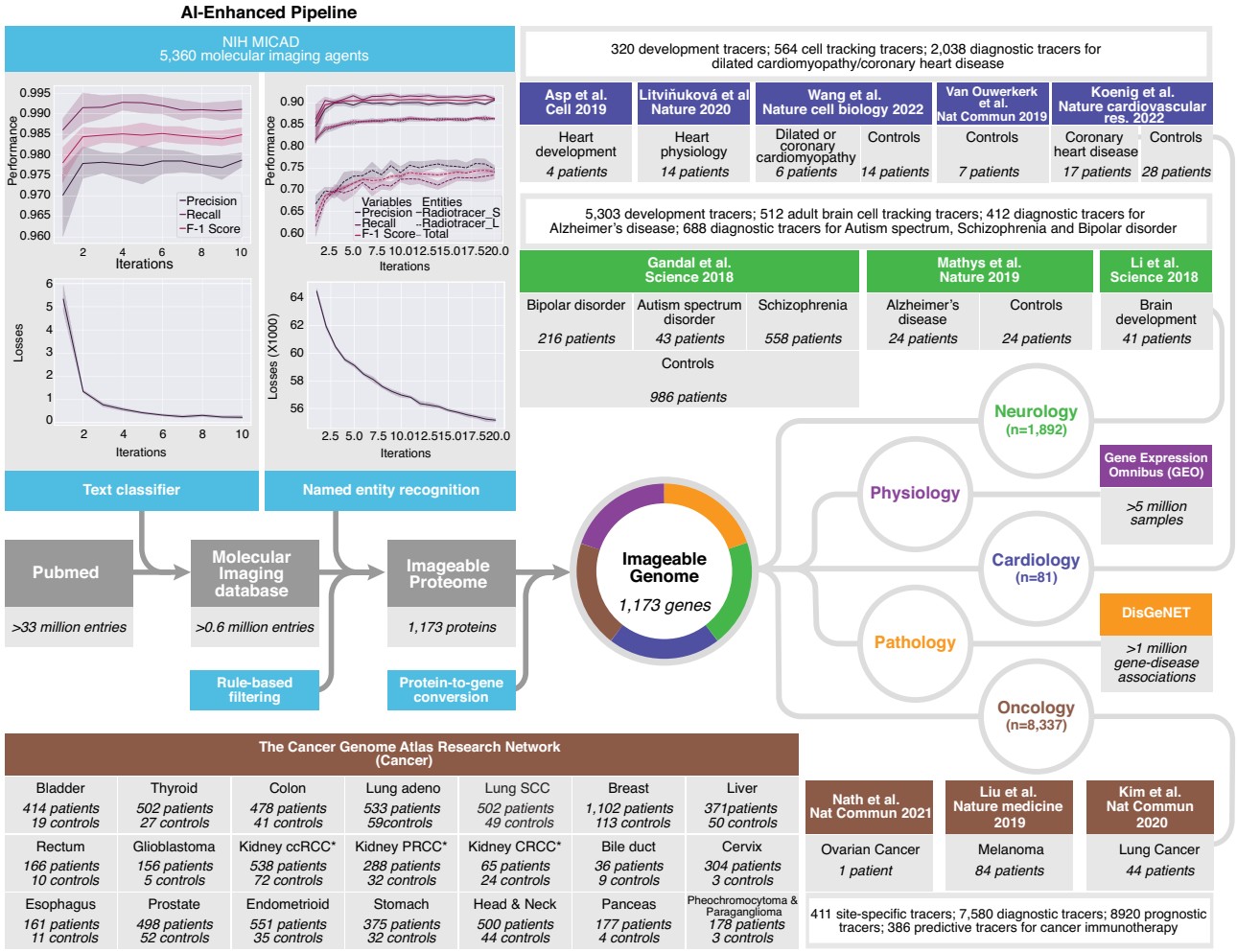

**Fig. 1 | Data pipeline.** For the *Imageable Genome* project, we developed a data pipeline that identifies texts containing radiotracers, recognizes and extracts names of radiotracers from texts, filters for clinically-relevant radiotracers and their associated targets, and translates protein names, i.e. of radiotracer targets, to names of the coding genes. We then downloaded the entire baseline MEDLINE/PubMed citation record, and used the above-mentioned pipeline to establish the *Imageable Genome*, the part of the human genome whose expression can be assessed by molecular imaging. Subsequently, we subjected the *Imageable Genome*

dataset to normal tissues expressions from a massive analysis of GEO studies, and gene-disease association from a curated DisGeNET database. Then we cross the *Imageable Genome* dataset to transcriptomic datasets of human brain development and disorders, heart development and failures, and The Cancer Genome Atlas (TCGA) of 21 cancer types, to identify imageable genes for monitoring tissue development, tracking cell types, as well as for diagnostic, prognostic and predictive imaging. ccRCC clear cell renal cell carcinoma, PRCC papillary renal cell carcinoma, CRCC chromophobe renal cell carcinoma.

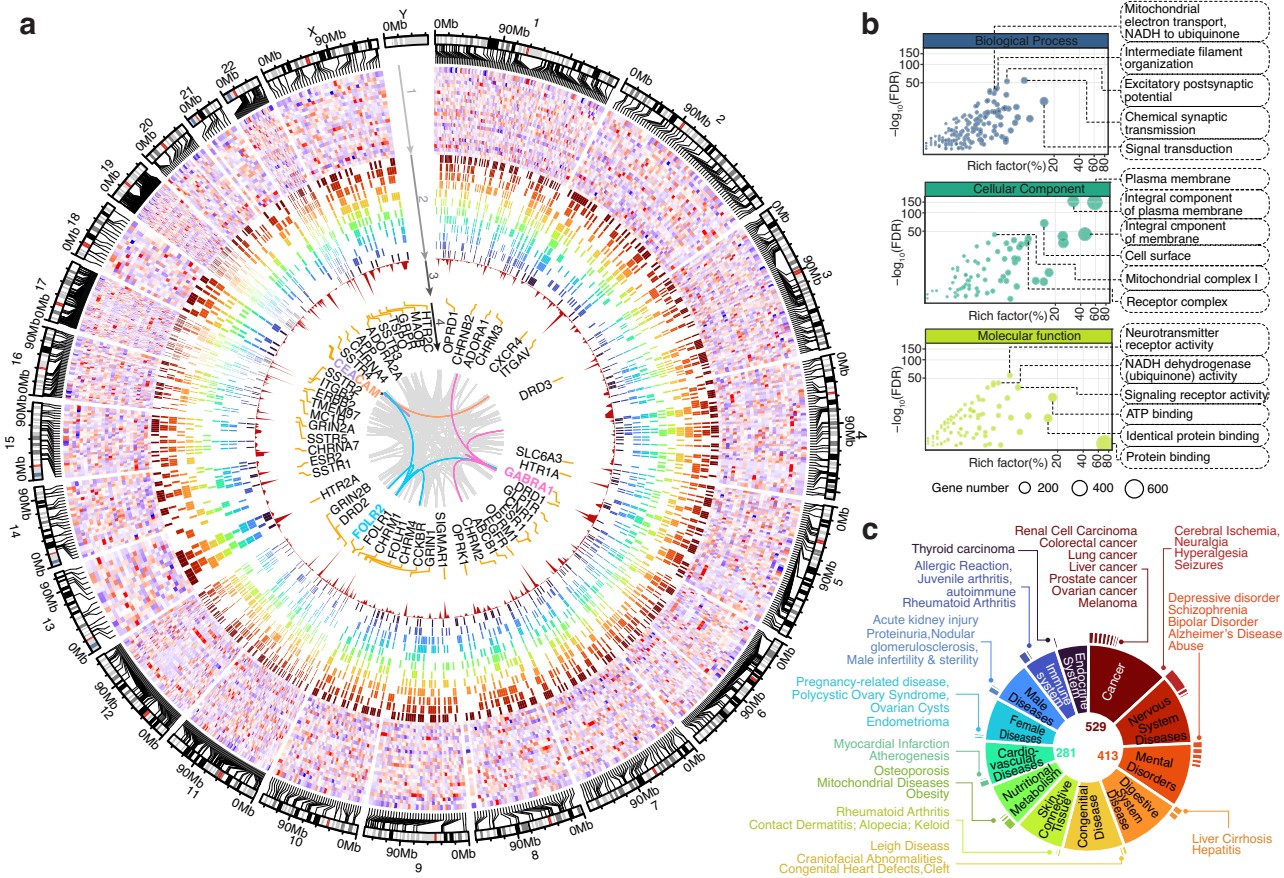

**Fig. 2 | The *Imageable Genome*. a** A circos plot depicting 1166 out of 1173 genes of the *Imageable Genome* with their chromosome locations (7 mitochondrial genes not shown): track 1, expression across 24 healthy tissues (red: relatively high gene expression, green: relatively low gene expression); track 2, gene-disease associations across 12 major diseases, with colour code corresponding to the disease classification in (**c**); track 3, number of radiotracers targeting an imageable gene. The height of red peak represents the corresponding number; track 4, imageable genes targeted by more than 30 radiotracers are labelled, and the links to their genome-wide co-expressed genes across 24 healthy tissues are highlighted in the innermost layer (pink, blue, purple and orange lines). CEACAM*: CEACAM3, CEACAM5, CEACAM6. **b** Scatter plot summarizing a list of enriched gene ontology (GO) terms from 1165 imageable genes (8 genes are not present in DAVID database). Rich factor is the ratio of the imageable gene number in a GO term to the total gene number. One-sided *p* values are computed using Fisher's Exact test with a 95% confidence interval. The False Discovery Rate (FDR) is used to control for multiple testing. GO terms passing the thresholds −log10(FDR) > 30 and Rich factor >15% are labelled. **c** Disease classification of 916 imageable genes. Top enriched 12 major diseases and their subcategorized diseases are shown in a sunburst plot, with the area corresponding to the ratio of gene number within a major disease to the total gene number. Centred absolute numbers represent the number of genes within the disease families cancer, mental disorders and cardiovascular diseases. Source data are provided as a Source data file.

chromosome (Fig. 2a), and 7 protein-coding genes on human mitochondrial DNA. It has a diverse expression pattern across different healthy organs[8–10] (Fig. 2a, track 1, Supplementary Data 3), and high occupancy in major human diseases[11], with most imageable genes associated with multiple diseases (Fig. 2a, track 2). Certain imageable genes are more frequently targeted (Fig. 2a, track 3), and show a correlative expression (Fig. 2a, track 4, labels in colour), indicating synergistic molecular imaging applications. Most imageable gene products are located on the cellular plasma membrane and function as signalling receptors (Fig. 2b). Recruiting the occupancy of each imageable gene via DisGeNET[11], we found that the most frequent disease domains include neurology, cardiology and oncology (Fig. 2c and Supplementary Fig. 2a–c).

Thus, by representing a key part of the human genome with both critical size and relevance, the expression of the *Imageable Genome* will inevitably be altered during the development and progression of any human disease. Conversely, the onset, progression and treatability of any human disease can potentially be assessed using the *Imageable Genome*. In the following sections, we confirm this hypothesis for a

wide spectrum of major neurologic, cardiologic and oncologic diseases.

## The *Imageable Genome* in neurology

The development of the human brain is a highly complex process that is controlled by precise spatiotemporal changes in genome expression[12,13], changes that significantly affect the expression of the *Imageable Genome*[14]. Twenty-nine percent of imageable genes are differentially expressed during brain development over the entire human life span (344 unique imageable genes, Fig. 3a, Supplementary Data 4 and Supplementary Fig. 3). 24% and 17% are differentially expressed in different regions of the prenatal and adult brain, respectively (286 and 202 imageable genes, Fig. 3b and Supplementary Data 5 and 6). The elucidation of imageable gene signatures in the brain over the human life span paves the way for molecular imaging to be used as a non-invasive monitoring tool for physiologic, and potentially pathologic, human brain development. To this end, the identified imageable gene signatures represent essential molecular benchmarks, synergistic to the recently established magnetic resonance imaging benchmarks for the developing brain[15].

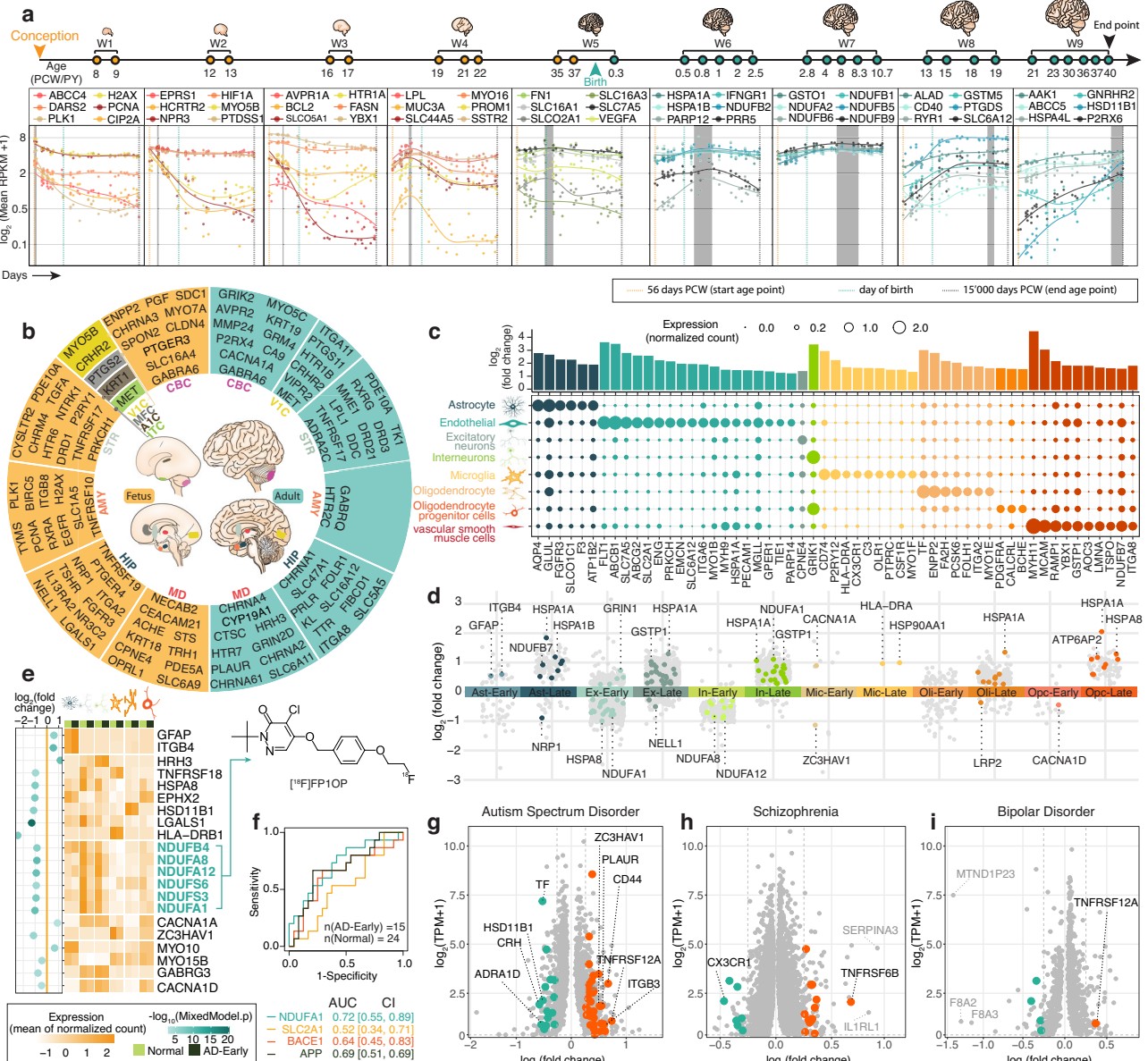

**Fig. 3 | The Imageable Genome in neurology. a** Representative temporally imageable genes across 9 brain development windows (W1–9). Dots represent the expression level calculated from all brain regions of a sample within a window. PCW: postconceptional weeks. PY postnatal years. Grey box: corresponding development window. **b** A donut chart showing the prenatal (left, yellow) and adult (right, lake blue) brain regional imageable genes. CBC, cerebellar cortex; V1C, primary visual (V1) cortex; STR, striatum; AMY, amygdala; HIP, hippocampus; MD, mediodorsal nucleus of thalamus; MFC, medial prefrontal cortex; A1C, primary auditory (A1) cortex; ITC, inferior temporal cortex. **c** Dot plot depicting the expression of cell type specific imageable genes in eight adult brain cell types, with colour code corresponding to cell type. **d** A scatter plot of Early or Late-Alzheimer's disease related imageable genes across 6 cell types. The top ranked imageable genes are highlighted with colour code corresponding to cell type. Ast astrocytes, Ex excitatory neurons, In inhibitory neurons, Mic microglia, Oli oligodendrocytes, Opc oligodendrocyte precursor cells. **e** Top ranked Early-Alzheimer's disease (AD-Early) related imageable genes sorted by absolute log2 (mean gene expression in AD-Early/mean gene expression in Normal) (shown as: log2(fold change)) values across 6 cell types. Z-score normalized read counts are shown (two sided Wilcoxon-rank sum test at FDR < 0.01, absolute log2(fold change) >0.25, and Poisson mixed model at FDR < 0.05). NADH:ubiquinone oxidoreductase supernumerary subunits were labelled in lake blue, followed by a chemical structure of Fluorine-18 radio-isotope labelled PET radiotracer targeting the mitochondrial complex. Cell-type icons created with BioRender.com. **f** Area under the receiver operating curve (AUC) values representing the capacity of imageable genes and 3 reference genes in discriminating Early-Alzheimer's disease and healthy brains. **g–i** Volcano plots of genes differentially expressed in **g** autism (ASD, n = 43), **h** schizophrenia (SCZ, n = 558), and **i** bipolar disorder (BD, n = 216) versus healthy brains (n = 986). Up-regulated (red) or down-regulated (green) imageable genes are highlighted. Ast astrocytes, Endo endothelial cells, Ex excitatory neurons, In inhibitory neurons, Mic microglia, Oli oligodendrocytes Opc oligodendrocyte precursor cells, Vsmc vascular smooth muscle cells. Source data are provided as a Source data file. Cell type icons are created with BioRender.com.

Moreover, the elucidation and comprehension of the complex organization of the developed human brain remains one of the most challenging endeavours in biomedical sciences. To understand differences in the expression of the *Imageable Genome* among human brain cells, we analysed 17,093 single-nucleus transcriptomes of 3 adult brains[12,13] and identified 55 imageable genes with a differential expression pattern among the 8 most prevalent brain cell types (Fig. 3c and Supplementary Data 7).

This global view of imageable gene signatures for human brain cells provides a framework to visualize and quantify the cellular

composition in different brain regions with molecular imaging, with the goal of understanding the healthy versus diseased adult brain. Notably, embryonic brains are composed of different major cell types, with a different profile of imageable genes (Supplementary Fig. 4).

The onset of neurodegenerative pathologies, such as Alzheimer's disease, alter the spatiotemporal expression of the *Imageable Genome*, a phenomenon which can be exploited to identify much needed methods for early disease detection. Analysing 80,660 single-nucleus transcriptomes from the prefrontal cortex of individuals with varying degrees of Alzheimer's disease pathology[16], we identified 41 cell-type-specific imageable genes up- or down-regulated exclusively in AD-early diseased versus Normal brains (Fig. 3d, Supplementary Fig. 5a and Supplementary Data 8), and 81 cell-type specific imageable gene up or down-regulated exclusively in AD-late diseased versus AD-early diseased brains (Supplementary Fig. 5a and Supplementary Data 8).

As a case in point, excitatory neurons and inter-neurons down-regulate genes encoding subunits of the NADH dehydrogenase early in the course of Alzheimer's disease (Fig. 3e), a phenomenon that can be imaged with [¹⁸F]FP1OP, originally developed for cardiac imaging[17]. [¹⁸F]FP1OP was, to the best of our knowledge, never tested in Alzheimer's disease; yet, receiver operating characteristic (ROC) analyses suggest a promising diagnostic accuracy (Fig. 3f).

Similarly, various neuropsychiatric conditions such as autism, schizophrenia and bipolar disorder are associated with an altered spatiotemporal expression of the *Imageable Genome*. Analysing bulk RNA-seq from 1695 brain samples[18], we identified 48, 21, and 5 imageable genes that are specifically expressed in autism (Fig. 3g), schizophrenia (Fig. 3h) or bipolar disorder respectively (Fig. 3i, Supplementary Fig. 5b and Supplementary Data 9). These results demonstrate how the *Imageable Genome*, by suggesting promising imaging targets, can help overcome the bottleneck of absent radiotracers for molecular imaging of neuropsychiatric disorders[19].

Finally, while the *Imageable Genome* suggests promising imaging targets for already existing radiotracers, it also guides target selection for novel radiotracers. We present 5940 stage-specific, 5897 region-specific, 535 cell-type-specific, and 1343 disease-specific genes that are likely more specific than those already present within the *Imageable Genome*, to image brain cells, Alzheimer's disease, autism, schizophrenia and bipolar disorder in Supplementary Data 10–15.

## The *Imageable Genome* in cardiology

The heart is the first organ to develop in the embryo; its development is orchestrated by specific spatiotemporal changes in genome expression[20,21], which significantly affect the expression of the *Imageable Genome*. Three percent of imageable genes are differentially expressed during embryonic development (41 imageable genes, Fig. 4a and Supplementary Data 16), whereas 4% and 9% are differentially expressed in different regions of the embryonic and adult heart, respectively (51 and 101 imageable genes, Fig. 4b and Supplementary Data 17 and 18). This comprehensive analysis of developmental gene signatures reveals a more stable expression of the *Imageable Genome* in the human heart than in the human brain, indicating that molecular imaging results for the human heart are less spatiotemporally sensitive. With the increasing understanding of the molecular mechanisms of congenital heart disease[22], the *Imageable Genome* envisages the potential role of molecular imaging as a non-invasive tool to monitor physiologic and pathologic heart development and provides a framework for its clinical implementation.

Moreover, elucidation of differences between the healthy and diseased human heart on a cellular level will to lead to more efficient ways of screening for, preventing, diagnosing and treating cardiac diseases. Such studies can be largely built on differences in the expression of the *Imageable Genome* among human cardiac cells. Analysing a comprehensive sc/snRNA-seq dataset from 6 different regions of 14 adult hearts, we identified 40 imageable genes

specifically expressed in each of the 11 cardiac cell types (Fig. 4c and Supplementary Data 19). Recent cardiac cell maps provided essential tools to deepen the understanding of the healthy[23] and the diseased heart[24]. The imageable gene signatures for human heart cells provide a tool to translate this understanding into clinically applied non-invasive mapping tools for cardiac cells.

Atrial fibrillation, coronary artery disease, and dilated cardiomyopathy each alter the expression of the *Imageable Genome* in cardiac cells, a phenomenon that can be exploited to identify much-needed methods for early disease detection[25]. By analysing bulk transcriptomes from cardiac tissue specimens of 7 individuals[26], we identified 6 imageable genes that are predictive for the onset of atrial fibrillation (Fig. 4d). Similarly, we identified 7 imageable genes characteristic for coronary artery disease by analysing single-nucleus transcriptomes from left ventricular tissue specimens of 2 patients and 14 healthy donors. The analysis of single-nucleus transcriptomes from left ventricular tissue specimens of 21 patients with dilated cardiomyopathy and 14 healthy donors[27,28] identified 278 imageable genes characteristic for dilated cardiomyopathy including the tetrameric protein haptoglobin, which is up-regulated by cardiac macrophages, endothelial cells, fibroblasts and smooth muscle cells (Fig. 4e–g, Supplementary Fig. 5c and Supplementary Data 20). Haptoglobin can be imaged with iodinated RM2-mab, originally developed for prostate cancer imaging[29]. RM2-mab was, to the best of our knowledge, never tested in dilated cardiomyopathy; yet, ROC analyses suggest a promising accuracy (Fig. 4h). These results demonstrate how specific disease-induced alterations in the expression of the *Imageable Genome* can be used to develop molecular imaging into a non-invasive tool for prediction and early detection of pertinent heart diseases.

To guide target selection for new and improved radiotracers for cardiac molecular imaging, we list 269 stage-specific, 1685 region-specific, 1356 disease-specific, and 752 cell-type-specific genes that are likely more specific than those already present within the *Imageable Genome*, to image heart development, heart cell organization, atrial fibrillation, dilated cardiomyopathy and coronary artery disease in Supplementary Data 21–25.

## The *Imageable Genome* in oncology

Genomic instability remains a major cause of resistance to cancer therapies[30]; it also induces spatiotemporal changes to the expression of the *Imageable Genome*. For instance, 5% of imageable genes in ovarian cancer are differently expressed at initial diagnosis, during therapy and in cases of relapse[31] (64 imageable genes, Fig. 5a, Supplementary Fig. 6a and Supplementary Data 26), whereas 10% of imageable genes are differently expressed in lung adenocarcinoma, compared to its advanced-stage primary tumour, pleural, bone and brain metastases[32] (117 imageable genes, Fig. 5b, Supplementary Fig. 6b and Supplementary Data 27). These spatiotemporal changes of *Imageable Genome* expression indicate that molecular imaging can become a tool to track shifts in genome expression during tumour progression.

Moreover, the expression of the *Imageable Genome* typically differs between human cancers and their tissue of origin. Overall, 3795 non-unique imageable genes (Supplementary Data 28) are up-regulated among 20 cancers of *The Cancer Genome Atlas*, including the glutamate metabotropic receptor GRM4 in breast cancer, the dopamine transporter SLC6A3 in renal clear cell cancer, the ionotropic glutamate receptor GRIN2D in colon cancer, the gamma-aminobutyric acid receptor GABRD in hepatocellular cancer and and the Epithelial Cell Adhesion Molecule EPCAM in cholangiocarcinoma (Fig. 5c(i) and Supplementary Fig. 7). The expression of these imageable genes is high in tumours, and low in tissues of origin and potential metastatic sites such as bone and liver (Fig. 5c(iii)). The respective radiotracers [¹⁸F]fluoromethyl-MK-801[33], [¹⁸F]FE-PE2I[34], [¹⁸F]fluoroethyl-

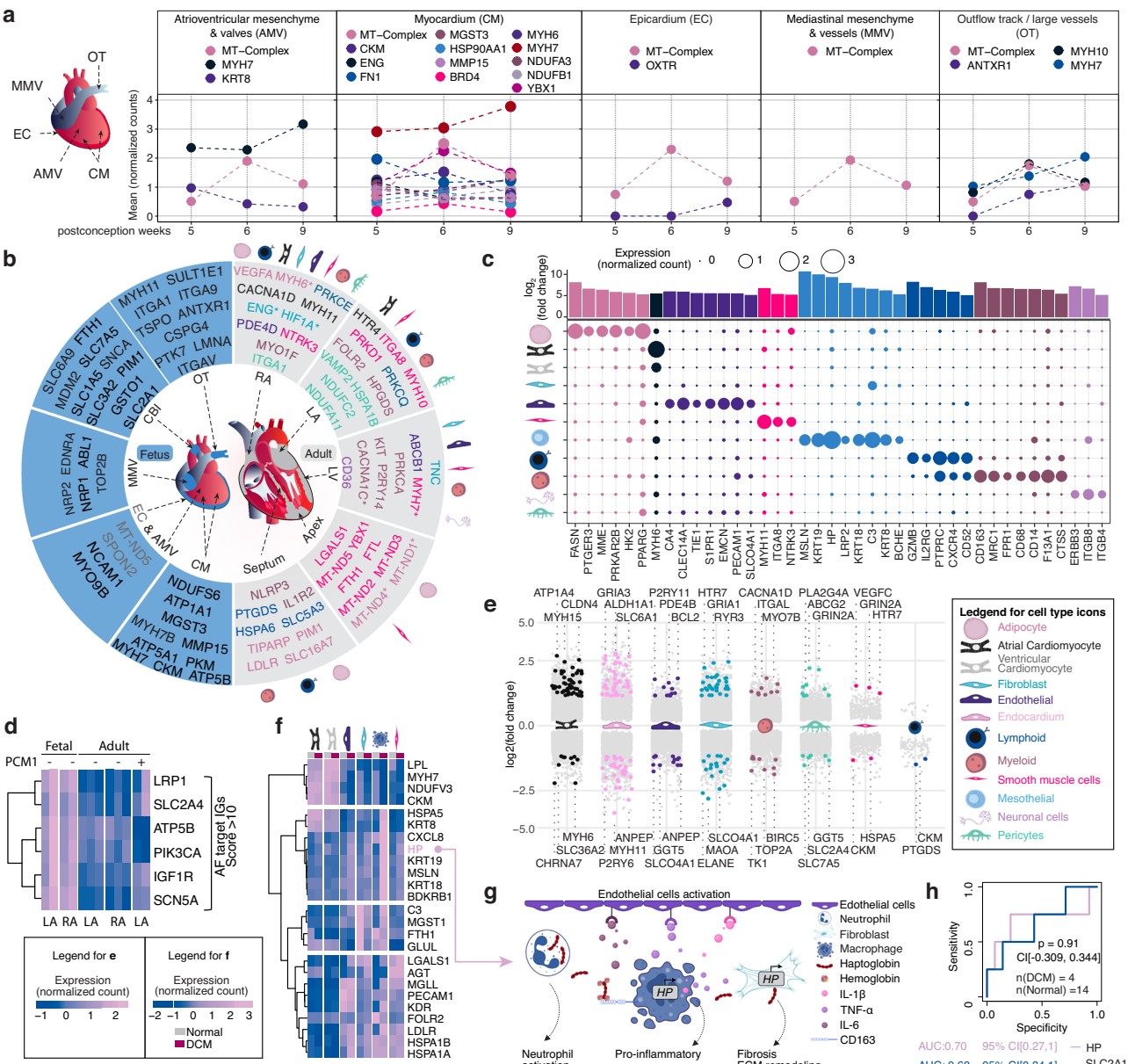

**Fig. 4 | The *Imageable Genome* in cardiology. a** Temporally imageable genes during human embryonic heart development within each region. A scheme showing the structure of the human embryonic heart at 5–9 post-conception weeks (Created with BioRender.com). Genes coding for mitochondria complex I-IV subunits are grouped and named into MT-complex. **b** Regional imageable genes for human embryonic heart (left, blue) and adult heart of each cell type (right, grey). CM cardiomyocyte, LV left ventricular, LA left atrial, RA right atrial, OT outflow track, CBI cavities with blood and immune cells, AMV atrioventricular mesenchyme and valves, MMV mediastinal mesenchyme and vessels, EC epicardium. **c** Dot plot depicting the expression of cell-type-specific imageable genes in eleven adult heart cell types, with colour code corresponding to cell type and circle size representing the relative gene expression level. **d** Heatmap of atrial fibrillation (AF) target imageable genes up- or down-regulated in adult PCM1 labelled (Pericentriolar Material 1 antibody) left atrial cells versus adult whole atrial tissue expression, or

up-regulated in the fetal atrium versus the adult atrium. **e** Expression of adult dilated cardiomyopathy (DCM) related genes in each cell type (DCM = 18, *n* (healthy) = 27). The top ranked imageable genes are highlighted in colour corresponding to cell type. **f** Heatmap of top ranked DCM related imageable genes in 6 cell types. Haptoglobin (HP) was labelled in pink. **g** Schematic illustrating the HP cellular function and potential pro-inflammatory effects under heart failure condition (Created with BioRender.com). **h** AUC values (95% confidence interval (CI)) representing the capacity of HP and GLUT1 (SLC2A1) expressions in discriminating DCM and healthy states of the adult human hearts including 4 cell types: EC (endocardium), FB (fibroblasts), MP (macrophages) and SMC (smooth muscle cells). *p* value with 95% CI calculated from two-sided DeLong's test for two ROC curves. Source data are provided as a Source data file. Cell-type icons are created with BioRender.com.

normemantine[35], [¹⁸F]flumazenil[36], 64Cu-DOTA-PEG-AntiEPCAM-Aptamer[37] and (Fig. 5c(ii)) were developed for neuroimaging and have never to the best of our knowledge been tested for tumour imaging; yet, ROC analyses suggest promising diagnostic accuracies (Fig. 5c(iv) and Supplementary Fig. 8a–j). These results indicate how cancer-specific expression profiles of the *Imageable Genome* can delineate new diagnostic tools for any cancer type.

The expression of the *Imageable Genome* also differs between cancers sensitive or resistant to specific therapies. For example, 47 imageable genes are differently expressed in melanomas sensitive or resistant to PD1-blockade[38] (in 34 cases with a *p* < 0.05, Fig. 5d and Supplementary Data 29), including the matrix metalloproteinase MMP9, the adenosine receptor ADORA1, the glycogen synthase kinase GSK3A, the folate receptor FOLR2 and the transforming growth factor

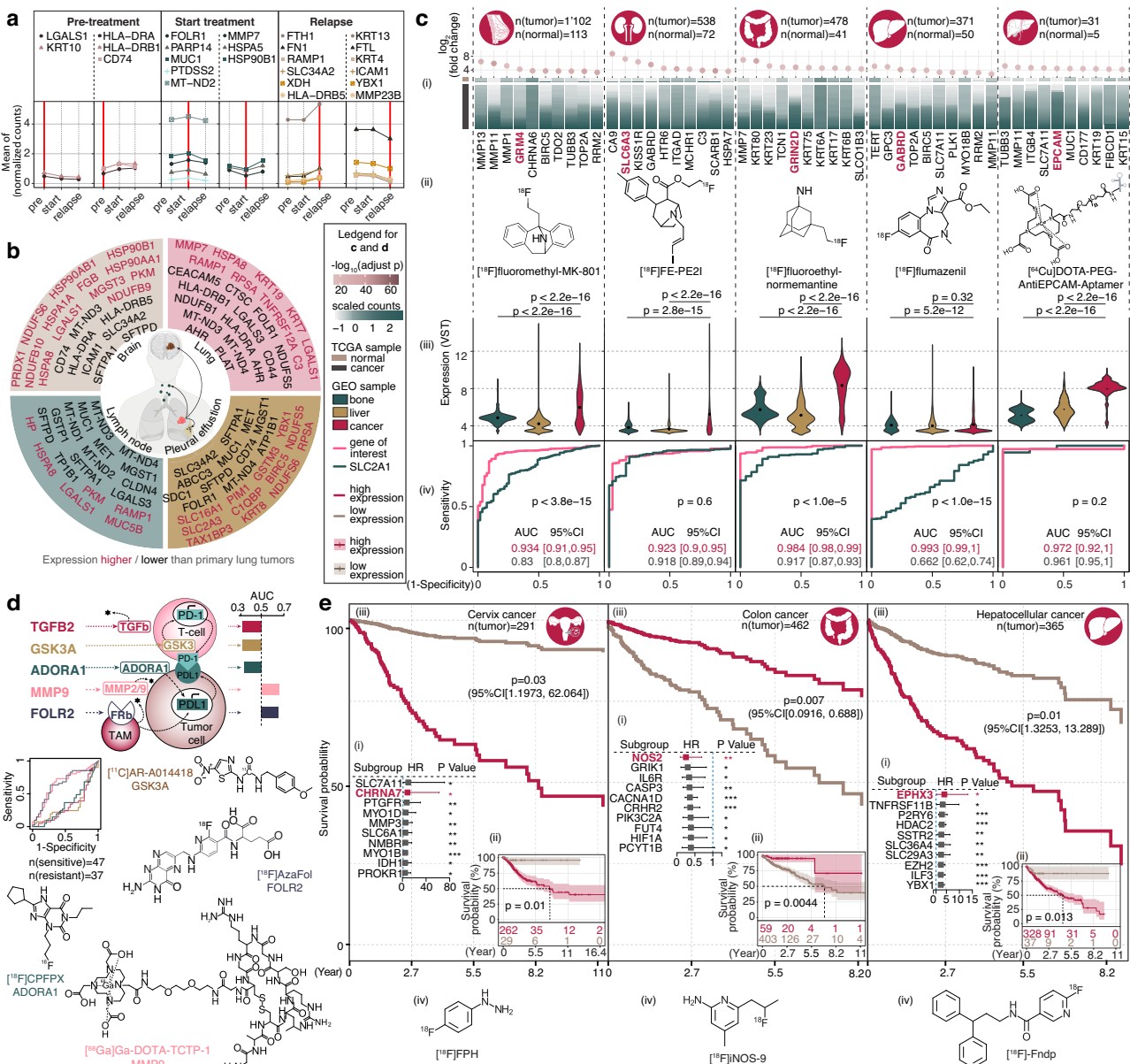

**Fig. 5 | The *Imageable Genome* in Oncology. a** Expression of top ranked disease status related imageable genes in malignant ovarian epithelial cells over time: at diagnosis (pre-treatment), under therapy (treatment), and in relapse as indicated by an augmented serum CA-125 level (two-sided Wilcox test). **b** Tissue origins during lung cancer metastasis and top ranked imageable genes differentially expressed at advanced or metastatic sites versus primary site, two-sided Student's *t* test $p < 0.01$ with Bonferroni correction<0.01, |log2(fold change)|>0.585. **c** Top diagnostic imageable genes across 5 cancer types. log2(fold change)>0.5 and Limma moderated t-statistic with Bonferroni correction<0.01. (i) heatmap illustrating the gene expressions (columns) in normal or cancer tissue (rows); (ii) an gene of interest (GOI) in mauve with a chemical structure of PET radiotracer targeting the GOI; (iii) violin plot of GOI expression in bone ($n = 284$), liver ($n = 1759$) and mammary cancer ($n = 5541$), kidney cancer ($n = 349$) or intestines cancer ($n = 2227$), two-sided Wilcoxon test $p$ values are shown. (iv) ROC curves of GOI and SLC2A1 with 95% confidence interval (CI) and two-sided DeLong's test $p$ value for

two correlated ROC curves. **d** A schematic showing the predictive capacity (ROC curves, sensitive: PD1-blockage treatment responder, resistant: non-responders) of 5 representative imageable genes, and their clinical radiotracers. **e** Prognostic imageable genes in, left to right, Cervix cancer, Colon cancer and Hepatocellular cancer. For each cancer, (i) top ranked imageable genes sorted by $p$ values from Wald test by fitting Cox proportional hazards (CPH) models to evaluate the effect of covariates on overall survival, with the best cutoff determined (see "Methods"). A square represents the Hazard Ratio (HR) with a horizontal line extending on either side representing the 95% CI. GOI is labelled in mauve. *$p < 0.05$; **$p < 0.01$; ***$p < 0.001$; (ii) Kaplan–Meier overall survival with Log-Rank test $p$ value; (iii) cox regression overall survival curves distinguishing GOI high/low expression groups (multivariable survival analyses with Wald test $p$ values by fitting CPH model); (iv) chemical structure of PET radiotracer targeting the GOI. Cancer-type icons are created with BioRender.com. Source data are provided as a Source data file.

TGFB2 (Fig. 5d). The respective radiotracers [68Ga]Ga-DOTA-TCTP-1[39], [18F]CPFPX[40], [11C]AR-A014418[41], [18F]AzaFol[42] and [89Zr]Zr-fresolimumab[43] were developed for diagnostic purposes. These results demonstrate how response-associated expression profiles of the *Imageable Genome* can delineate new predictive tools for cancer therapies.

The expression of the *Imageable Genome* also correlates with cancer survival. The expression of 6831 imageable genes differs in various cancer types and their tissue of origin, and correlates with overall survival in univariate and multivariate analyses (Supplementary Data 30), including the Cholinergic Receptor CHRNA7 in cervix cancer, the Nitric Oxide Synthase NOS2 in colon cancer, and the Epoxide

Hydrolase EPHX3 in Hepatocellular cancer (Fig. 5e(i, ii)). The respective radiotracers [18F]FPH[44], [18F]iNOS-9[45] and [18F]-Fndp[46] were developed for brain and heart imaging (Fig. 5e(iv)). They were, to the best of our knowledge, never used for cancer imaging; yet, survival analyses suggest promising prognostic potential (Fig. 5e(iii) and Supplementary Fig. 9). These results demonstrate how survival-associated expression profiles of the *Imageable Genome* can delineate new prognostic tools for any cancer type.

Finally, we list 4127, 924 and 5583 genes that are likely more specific than those already present within the *Imageable Genome* for diagnostic, predictive and prognostic cancer imaging, as well as prognostic applications for the most frequently used tracers in the clinic today in Supplementary Data 31–37.

### The *Imageable Genome* in COVID-19

Diagnosing COVID-19, identifying affected tissues, and understanding its impact on human health remain a global health care challenge. Analysing 106,792 and 40,880, single-nucleus transcriptomes of SARS-CoV-2 infected lung and heart[47], we identified 36 cell-type specific imageable genes up or down-regulated exclusively in COVID-19 versus healthy tissues[48] (Supplementary Fig. 10a–c). These results demonstrate how the *Imageable Genome* might allow expanding molecular imaging beyond neurology, cardiology, and oncology into new fields.

## Discussion

The *Imageable Genome* represents a novel and timely means of approaching, summarizing and understanding the field of PET-based molecular imaging. In bridging medical imaging, genomics, systems biology and data science on a broad scale to yield easily accessible results that are of high interdisciplinary interest and relevance, the approach underpinning the *Imageable Genome* is truly unique.

The *Imageable Genome* amounts to 1.8% of the human genome and 6% of the human protein coding genome[49]. It represents a part of the human genome that has the critical size and focus to be altered during development and progression of any human disease. So far, about 1–30% of human protein coding genes have been identified as disease related markers in neurology, cardiology and oncology, among which 21% are imageable genes (Supplementary Fig. 11). This implies that the transformation of healthy tissues into diseased tissues, the development of treatment-resistance and the development of traits worsening the overall prognosis most likely affect the expression of the *Imageable Genome*. Thus, the *Imageable Genome* will likely facilitate the development of diagnostic, predictive and prognostic imaging tool for any human disease.

The *Imageable Genome* is constantly growing, and its relevance is likely to grow correspondingly. The availability of more single cell sequencing data in the future might allow to identify even more imageable genes with relevance in human diseases. This dynamic growth combined with the ability to image targets of high pertinence among a wide range of human diseases represent major *strengths* of the field of molecular imaging. While the low clinical translation rate of the thousands of radiotracers targeting the *Imageable Genome* represents an important *weakness*[50], the ability of the *Imageable Genome* to bridge genomics with molecular imaging represents a major *opportunity* for PET-based molecular imaging to become an established means of studying systems biology as well as a clinical gateway for molecular medicine.

Since its introduction over 70 years ago with Linus Pauling's seminal paper on sickle cell anaemia[51], the field molecular medicine has elucidated the origins of many human diseases on a molecular level and translated this understanding into tools for disease prevention, diagnosis, prognosis and treatment. Nearly 20 years ago, the *Human Genome Project* significantly empowered research into the genetic basis of human disease, and amplified the potential for molecular discoveries that could be translated into clinical tests[52,53]. However,

during the last 10 years, the translation rate of genomic discoveries into FDA-regulated tests dropped from approximately 0.01%[54–56] to 0.001%[55–57]. A fundamental challenge in operationalizing genomic discoveries remains the need to decipher the code of complex research-driven multi-omics discoveries into the simple decision-oriented language of the clinic. The *Imageable Genome* can serve as a "Rosetta stone" that systematically translates these complex genomic discoveries into easily applicable clinical imaging tests.

Translating genomic discoveries into molecular imaging tests will open new avenues for molecular medicine. Sample collection in certain organs, such as the brain or heart, represents a well-known barrier to genomic analysis. Non-invasive PET-based molecular imaging as a surrogate for genomic analyses could overcome this barrier, permitting "sample-free genomics". Moreover, the necessity for repeated sampling of diseased tissue remains a relative barrier to recurring genomic analysis. PET-based molecular imaging could similarly overcome this with non-invasive "serial genomic testing". Finally, suboptimal or non-representative sampling remains a key limitation in genomic analyses of heterogeneous diseases such as systemic inflammatory diseases or advanced cancers. The use of PET-based molecular imaging could potentially overcome this barrier with "whole-body genomics" by providing information not only on the primary lesion, but all lesions in a patient with widespread disease.

Molecular imaging, and especially PET, was originally envisioned as a clinical tool to assess, visualize and quantify systems biology in vivo[2]. However, during the last decade PET, in the form of FDG PET-CT, became mainly an increasingly sensitive diagnostic tool used in the setting of cancer diagnosis, staging and restaging. The combination of the genomic and AI revolutions now paves the way for the initial vision of PET imaging to be fully realized. First, the rise of systems biology has provided an consolidative understanding of complex biological systems, many of which can be imaged by PET[58]. Second, the introduction of high-throughput techniques, particularly transcriptomics, has generated the necessary data to model these biological systems[59]. Third, the widespread availability of low-cost, high-throughput transcriptomics has created a wealth of transcriptomic data[60], which will fuel the research necessary to establish PET as a systems biology tool. Finally, the rise of AI now provides the means to obtain a global view on the entire field of molecular imaging in the example of the *Imageable Genome*, which finally allows bridging the field with transcriptomic data on a major scale.

By bridging genomics with the entire field of PET-based molecular imaging, completely new avenues for molecular imaging will be opened. With the examples of cerebral and cardiac development and tissue composition, we demonstrate how the use of genomics can establish PET-based molecular imaging as a tracking tool for organogenesis and distinct cell populations. For example, with availability of the necessary genomics data, one could envision detailed, serial in-vivo imaging studies of developmental pathologies such as Tetralogy of Fallot or cerebral neuronal migration disorders performed in embryonic animal models with PET tracers identified by the *Imageable Genome*. Such studies could potentially elucidate the spatiotemporal mechanisms underlying these diseases on a molecular and cellular level, as well as possibly identify a means of treatment. With the examples of atrial fibrillation and PD1-blockade in oncology, we demonstrate how PET can be tailored to detect patients that are prone to develop a specific disease, or identify diseases that will respond to a specific therapy. Such applications of molecular imaging can be directed by genomic discoveries, and the use of modern automated radiopharmacy systems[61] will make the necessary variety of radiotracers available to create "genomic imaging centres".

A widely available *Imageable Genome* will likely have a high interdisciplinary impact on multiple branches of the life sciences. First, our approach of combining traditional systematic review techniques with machine learning and deep learning algorithms to meta-analyse

entire clinical fields has unlimited potential. For example, if repeated to compile and meta-analyse the entire field of immunohistochemistry, the entire randomized evidence in oncology or the entirety of toxicities reported for specific drugs, medicine would enter a new era of truly evidence-based, informed clinical decision-making. Yet, such meta-analyses will increasingly rely on the validity of claims within the compiled literature and on the subsequent validation analyses of the outcome of these human-AI pipelines. New challenges such as replicability issues will raise as a consequence of the magnitude of these results and the lack of external validation datasets, and they will require the collaborative efforts of the scientific community and novel external proofing techniques.

Second, our approach highlights a very timely core value: sustainability. Reducing the costs and the carbon fingerprint associated with scientific research is crucial[62], and we see the *Imageable Genome* as a prime example of "sustainable science" that facilitates the recycling and re-purposing of previously developed radiotracers on a large scale, and moreover, reviving the initial vision for PET, molecular imaging and molecular medicine.

Third, several manually compiled databases might benefit from the *Imageable Genome*, which was generated using an AI-based approach. One example is the NIH MICAD, which can now be updated and significantly expanded through the *Imageable Genome*. Another example is the *Druggable Genome*, which currently includes 667 genes whose products can be targeted by drugs for treatment of human disease[63]. As PET-based molecular imaging has an established role in identifying, screening and validating new drug targets[64], we expect a high translation of imageable genes into druggable genes, hence a growth of the *Druggable Genome* catalysed by the *Imageable Genome*.

Fourth, we hope that the results from our approach of cross-referencing the *Imageable Genome* with key datasets from neurology, cardiology and oncology will spark interest from clinicians and researchers in these fields. Interestingly, we found a high cross-usability of radiotracers among the three clinical fields, which is in line with the growing evidence of impressive overlap among genes responsible for wide-ranging pathologies, such as those driving cancer as well as underlying certain development disorders[65,66]. A "big science" approach with an interdisciplinary collaboration among researchers, clinicians, geneticists and medical imagers will be essential to translate the findings from the *Imageable Genome* into clinical use.

Finally, academic and commercial radio-pharmacological developments are likely to be vastly impacted by advances in knowledge put forth by the *Imageable Genome*. By identifying genes of high clinical pertinence that are not yet imageable, the *Imageable Genome* will be a key resource to guide the development of a new generation of improved radiopharmaceuticals, which will in turn further expand the *Imageable Genome*.

In describing the *Imageable Genome*, we provide a global view of molecular imaging and demonstrate how this field is only at the beginning of realizing its potential to bring molecular medicine fully into the clinical realm. We hope that the *Imageable Genome* will serve as a key resource to catalyse further high-yield research, and to advance the impact of molecular medicine on human health.

## Methods

For the *Imageable Genome* project, we developed a data pipeline that identifies texts containing radiotracers, recognizes and extracts names of radiotracers from texts, filters for clinically relevant radiotracers and their associated targets, and translates protein names, i.e. of radiotracer targets, to names of the coding genes. We then downloaded the entire baseline MEDLINE/PubMed citation record, and used the data pipeline to establish the part of the human genome whose expression can be assessed by molecular imaging. Subsequently, we subjected the

dataset to normal tissues expressions from a massive analysis of GEO studies, gene ontology, and gene-disease association from a curated DisGeNET database. Then we crossed the dataset to transcriptomic datasets of human brain development and disorders, heart development and disorders, and The Cancer Genome Atlas (TCGA) of 20 cancer types, to identify novel development-tracking, cell type specific, diagnostic, prognostic and predictive imageable genes.

### Text classifier

We developed a text classifier to identify texts containing a radiotracer for molecular imaging. In brief, on February 26, 2022 we downloaded the entire baseline MEDLINE/PubMed citation record, and created a dataset of 33,405,863 PubMed ID (PMID) citations and 22,542,347 abstracts within 1114 xml files[67,68]. Using PubMed_Parser[69], we parsed these xml files into a series of Pandas.DataFrames[70] that we saved to 90 parquet files. To establish a ground truth of texts containing a radiotracer for molecular imaging, we downloaded the entire Molecular Imaging and Contrast Agent Database (MICAD)[71,72], containing 5360 molecular imaging agents, their imaging modality, PMID, abbreviated name, chemical name, application, and molecular target, as an xlsx file. From this file, we extracted all radiotracers for single photon emission computed tomography (SPECT) and positron emission tomography (PET), and saved them into a python dictionary with their respective PMIDs, resulting in 3550 entries. From these entries, we produced a list of PMIDs and filtered out the duplicates obtaining 2997 unique PMIDs.

We used these 2997 PMIDs to search for the corresponding PubMed citations in the parquet files, and obtained a list of 2060 abstracts containing a radiotracer for molecular imaging. To these, we added 2308 random abstracts from PubMed not containing radiotracers. We verified the correct labelling of all abstracts by a team of experts in the field using the collaborative annotation platform Doccano[73], and generated a training corpus of 4368 annotated abstract texts. Prior to the model training, we performed pre-processing by removing any leading and trailing spaces, making all the characters lowercase and eliminating all punctuation signs excepting brackets, dashes and percentage symbols.

We then performed a 75/25% train/test split of this training corpus and trained a text categorizer model consisting of a convolutional neural network using python's natural language processing tool spaCy[74], and the pre-trained model "en_core_sci_md" from ScispaCy[75], chosen for its efficiency, its state-of-the-art performance, and its compatibility with spaCy. We trained the model 10 times over 10 iterations to classify abstract texts describing a radiotracer with an average precision of 97.8%, an average recall of 98.8%, and an average F1 score of 98.3% (with a set prediction score threshold of 0.5, being the numerical value given by the softmax function in the output layer of the model to the "positive" category).

### Named entity recognition

We developed an algorithm to recognize and extract names of radiotracers within texts. We used the 2060 abstract texts describing radiotracers to produce a training corpus for a named entity recognition model. We used the python tool FuzzyMatcher from Spaczz[76] to search for matches of a list of the abbreviated names of radiotracers from MICAD within the 2060 abstract texts. We then turned these matches into entity labels ("RADIOTRACER") that were verified by a team of experts in the field using Doccano[73]. We trained a named entity recognition model consisting of a convolutional neural network using python's natural language processing tool spaCy[74], and the pre-trained model "en_ner_jnlpba_md" from ScispaCy[75]. This pre-trained model detects DNA, CELL_TYPE, CELL_LINE, RNA and PROTEIN with an F1 Score of 70.9%. To avoid a "catastrophic forgetting" event[77], we used this model on each sentence of each abstract, turned the matches of the model known entities into entity labels for a new training step, and added the previously annotated "RADIOTRACER" labels.

We then divided the "RADIOTRACER" label into two new labels according to the token length of the entities: RADIOTRACER-L for entities with more than 1 token length, and RADIOTRACER-S for entities of exactly one token length in order to improve performance of the model by creating two more homogeneous entities. The training dataset consisted of 13,863 sentences with a total of 38,194 annotated entities (RADIOTRACER 15,324, PROTEIN: 7003, DNA: 1104, CELL_- TYPE: 840). We re-updated the model with these new entities over 20 iterations obtaining a precision of 76.7%, a recall of 75.0% and an F1 score of 75.9% for RADIOTRACER-L, as well as a precision of 90.1%, a recall of 90.7%, and an F1 score of 90.4% for RADIOTRACER-S.

**Rule-based filtering**
We developed an SQL search algorithm that allows filtering for clinically used radiotracers and radiotracer-to-target associations. Specifically, we generated a set of numeric sequences corresponding to the mass number of radioisotopes used for molecular imaging[78], and an algorithm to retain only those texts in which the radiotracers contained one of these numeric sequences. Furthermore, we developed a filtering algorithm to retain texts that contain both, a radiotracer (RADIOTRACER-L and/or RADIOTRACER-S) and a protein or gene name in the title and/or the abstract.

**Protein-to-gene translation**
We developed a tool that can translate protein names in our final excel file exported from the SQL database after the rule-based filtering step, i.e. those of radiotracer targets, to the names of the respective coding genes. Thereby, we used mygene, the python wrapper for the mygene.info REST API[79], a database which contains large lists of protein aliases and their coding genes. We wrote an automatic querying algorithm that iterated over all target proteins, wrote the best match to the table provided by mygene.info (matched protein, associated gene symbol and associated gene ensembl ID) and exported it to a new excel file for manual verification.

**The *Imageable Genome***
We downloaded and parsed the entire baseline MEDLINE/PubMed citation record, and created a dataset of 33,405,863 PubMed ID (PMID) citations and 22,542,347 abstracts. Using the text classifier, we identified within this dataset 649,995 abstracts with a prediction score above 0.5 for containing a radiotracer. We stored these to an SQL partitioned database hosted on a MYSQL server[80] containing all original columns of the baseline dataset plus a column with the prediction score. We then used the named entity recognition model to extract radiotracers from all title and abstract texts in the dataset, and added that information to the SQL partitioned database. We then used our filter algorithms, and retained all entries that contained both, a radiotracer and a protein or gene name in the title and/or the abstract, and retained all entries in which the radiotracers contained a numeric sequence of a clinically relevant radiotracer. Using the rule-based filtering, we obtained a dataset of 44,811 entries that we stored to an SQL partitioned table.

Finally, a team of experts in the field verified the correct identification of the radiotracer, the correct identification of the gene entity, the correct identification of the protein entity, the correct association between the radiotracer and the gene entity, the correct identification of the protein entity, the correct association between the radiotracer and the protein entity. Unclear cases were solved by a second or third reviewer.

**The Imageable Genome in Gene Expression Omnibus (GEO) human normal tissue RNA-seq datasets[9,10]**
We used a recently published re-processed RNA-seq data source comprising Variance Stabilizing Transformation normalized counts across 238,522 human samples generated from Illumina platforms with corresponding entries in the GEO database (ARCHS4 v8, February 2020). We used "correlationAnalyzeR" (R package, Version 1.0.0) to retrieve normalized and transformed RNA-seq Variance Stabilizing Transformation data (function "getTissueVST") and correlation matrix (Function "analyzeSingleGenes") from Azure MySQL server as described in the paper. We included a total of 48,619 samples from 24 human normal tissues and 1161 *Imageable Genome* genes (HSP90AA3P, MMP26, MUC19, KRT33A, KRT28, MC3R, MIR155, GSTA5, HLA-DRB9, GSTA6P, OR12D1, CYP11B2, and their alias are not present in MySQL dataset) in this section.

**Gene ontology (GO) of the *Imageable Genome***
We used DAVID Bioinformatics Resources (2021 Update)[81,82] to perform gene ontology functional enrichment analysis for all 1173 *Imageable Genome* genes.

**Disease enrichment analysis for the *Imageable Genome***
We downloaded from DisGeNET CURATED database (V7.0) the curated gene-disease association information which contains gene-disease associations from UNIPROT, CGI, ClinGen, Genomics England, CTD (human subset), PsyGeNET, and Orphanet. We used "disgenet2r" (R package, version 0.99.2) with function "disease_enrichment" to retrieve disease enrichment information over 24 human diseases and 916 imageable genes[11], focused on genes with the DisGeNET score ≥0.3. As described by the DisGeNET, the DisGeNET score was determined by the Gene-Disease Association (GDA) scores, which takes into account the number and type of sources (level of curation, organisms), and the number of publications supporting the association. Disease Specificity Index (DSI) and Disease Pleiotropy Index (DPI) for each gene was also retrieved and used for the implementation of disease associations.

**The *Imageable Genome* in brain development[14]**
We downloaded bulk RNA-seq count matrix from the National Institutes of Health–funded PsychENCODE (http://psychencode.org). The dataset included 607 samples, 16 anatomical brain regions from 41 post-mortem individuals, with ages ranged from 8 post-conception weeks (PCW) to 40 postnatal years (PY) (Window 1–9). Regions and sequencing data collection were as described in this article[12]. In brief,

- Neocortex (NCX), including Frontal cortex: orbital (OFC), dorsolateral (DFC, aka DLPFC), ventrolateral (VFC), and medial (MFC), prospective motor and parietal somatosensory (MSC) cerebral wall, Orbital prefrontal cortex (OFC), Dorsolateral prefrontal cortex (DFC), Ventrolateral prefrontal cortex (VFC), Medial prefrontal cortex (MFC); Parietal cortex: prospective inferior parietal cortex (IPC), Primary somatosensory cortex (S1C); temporal cortex: auditory and superior temporal cortex (A1C/STC) cerebral wall, Posterior superior temporal cortex (STC), Inferior temporal cortex (ITC) and Occipital cortex: primary visual cortex (V1C).
- Hippocampus (HIP)
- Amygdala (AMY)
- Striatum (STR)
- Mediodorsal nucleus of thalamus (MD)
- Cerebellar cortex (CBC).

We performed differential gene expression analysis as described in this article[12]. In brief, we computed temporal differentially expressed genes for each pair of in-window and out-window samples (pairwise windows comparison) across all regions. We also computed regional differentially expressed genes for each pair of in-region and out-region samples (pairwise regions comparison) across all windows.

We used DESeq2 (R package, version 1.32.0) to perform differential gene expression analysis, using bulk RNA-seq count matrix as input and the two sequencing sites (Yale and USC) as covariates to reduce batch effect. Genes from chrMT were excluded from the analysis. We defined the differentially expressed genes by passing the

filtering criteria: count >10 in at least one condition, log2(fold change) >1, mean RPKM > 1 for the case condition and false discovery rate <0.01. We then crossed the lists of temporal or regional differentially expressed genes to the *Imageable Genome* to identify imageable markers for brain development. From Charles's study (Cell, 2022), we analysed snRNA-seq data of 26 post-mortem prefrontal cortex (PFC) samples from individuals spanning foetal, neonatal, infancy, childhood, adolescence, and adult stages of development and retrieved 14,984 unique development-associated differentially expressed genes (devDEGs, false discovery rate [FDR] <0.05) within at least one major trajectory (cell type). We used the list of devDEGs as a validation dataset crossing to the imageable brain development markers obtained from Li's study.

### The *Imageable Genome* in human brain cells[12]
We downloaded single cell RNA-Seq (scRNA-seq) count matrix and single nucleus RNA-seq (snRNA-seq) count matrix from the National Institutes of Health–funded PsychENCODE (http://psychencode.org). The scRNA-Seq dataset included 60,155 genes and 762 single cells from 8 prenatal donors. Cells were classified into 6 major cell types: ExN: excitatory neurons, InN: interneurons, IPC: intermediate progenitor cells, NasN: nascent neurons, OPC: oligodendrocyte. The snRNA-Seq dataset included 23,476 genes and 17,093 single nuclei from 3 adult donors. Cells were classified into 8 major cell types: Astro: cells in the astroglial lineage, Endo: endothelial cells, ExN: excitatory neurons, InN: interneurons, Microglia, Oligo: oligodendrocytes, OPC: oligodendrocyte progenitor cell, VSMC: vascular smooth muscle cells.

Genes from chrMT were excluded from the analysis. From "Seurat" (R package, version 4.0.5), we used a global-scaling normalization function "NormalizeData" (normalization.method = "LogNormalize", scale factor = 10,000) to normalized the raw counts, and function "FindAllMarkers" to identify marker genes for each major cell type.

To identify marker genes with a specificity for a given cell type, we compared each group of cells to the rest of the cells and defined cell type specific markers by passing the filtering criteria: for each comparison, the percentage of cells where the gene is detected in the group >30%, the percentage of cells where the gene is detected outside the group <30%, a difference between percent of cells expressing gene within and outside group >30%, a false discovery rate <0.00001, and a log2(fold change) >1. We then crossed the list of markers to the *Imageable Genome* to identify brain cell type specific imageable markers during brain development.

### The *Imageable Genome* in Alzheimer's disease[16]
We retrieved cell type markers for either early Alzheimer's disease versus control, or late Alzheimer's disease versus early Alzheimer's disease from the study's Supplementary Tables 1–3. We downloaded snRNA-seq count matrix from AD Knowledge Portal (Synapse ID: syn18485175). This dataset included 17,926 genes and 70,634 nuclei from the prefrontal cortex of 48 individuals (15 individuals classified into "early-pathology", 9 individuals classified into "late-pathology", and 24 individuals classified into "no-pathology"). Cells were classified into 8 major cell types: Ast: astrocytes, End: endothelial cell, Ex: Excitatory neurons, In: Inhibitory neurons, Mic: microglia, Oli: Oligodendrocytes, Opc: Oligodendrocyte precursor cells, Per: pericytes. From "Seurat" (R package, version 4.0.5), we employed a global-scaling function "NormalizeData" (normalization.method = "LogNormalize", scale factor = 10,000) to normalized the raw counts. We define cell type markers by passing the filtering criteria: TRUE in both modules and mean log transformed count within case >0.3. We then crossed the list of early Alzheimer's disease and late Alzheimer's disease cell type markers to the *Imageable Genome* to identify cell type specific imageable markers for Alzheimer's disease.

### The *Imageable Genome* in autism spectrum disorder, schizophrenia, and bipolar disorder[18]
We retrieved differentially expressed genes for autism spectrum disorder, schizophrenia, and bipolar disorder from the study's Supplementary Table S1. We downloaded bulk RNA-seq TPM matrix and clinical metadata from PsychENCODE Consortium (http://resource.psychencode.org/). This dataset included 57,820 genes and 1867 subjects with their clinical status. We included a total of 1803 subjects diagnosed with autism spectrum disorder ($n = 43$), schizophrenia ($n = 558$), or bipolar disorder ($n = 216$), and controls ($n = 986$) for the downstream analysis. We define differentially expressed genes by passing the filtering criteria: false discovery rate <0.05, |log2(fold change)| > 0.25, mean TPM in the case group >0.3 for up-regulated genes and mean TPM in the control group >0.3 for down-regulated genes. We then crossed the lists of differentially expressed genes to the *Imageable Genome* to identify imageable markers for autism spectrum disorder, schizophrenia, and bipolar disorder.

### The *Imageable Genome* in heart development[20]
We downloaded filtered Spatiotemporal count matrix and the meta table from https://www.spatialresearch.org/resources-published-datasets/doi-10-1016-j-cell-2019-11-025/. This dataset included 39,739 genes and 3111 spot samples from heart specimens at 5-, 6- and 9-week development. Following the article's methods, we annotated the gene symbols using ENSEMBL genome assembly GRCh38, release 86 (gencode.v25.basic.annotation.gff3), kept only protein coding and lincRNAs, removed MALAT1 and MTRNR genes as well as highly expressed genes related to haemoglobin and linked to the Y-chromosome, excluded spots with fewer than 500 genes, and excluded genes expressed in fewer than 15 spots. We finally obtained 14,002 genes for downstream analysis.

From "Seurat" (R package, version 4.0.5), we used the function "NormalizeData" (normalization.method = "LogNormalize") to normalize the filtered raw counts, by setting the scale factor with the average of column sums across the expression matrix.

To compute the regional markers, we firstly redefined the regions. There were 10 clusters annotated in the article's original meta file: compact ventricular myocardium (cluster0), trabecular ventricular myocardium (cluster1), trabecular ventricular myocardium (cluster2), trabecular ventricular myocardium (cluster3), atrial ventricular myocardium (cluster4), outflow tract/larger vessels (cluster 5), atrioventricular mesenchyme and valves (cluster 6), mediastinal mesenchyme and vessels (cluster 7), cavities with blood and immune cells (cluster 8), and epicardium (cluster 9). We combined the samples in clusters 0–4 to obtain a global region group renamed into "myocardium", and kept the clusters 5–9.

We then computed regional markers for each pair of in-region and out-region samples (pairwise regions comparison) across all timepoints. We also computed temporal-regional markers for each pair of in-timepoint and out-timepoint samples (pairwise temporal comparison) for each region.

From "Seurat" (R package, version 4.0.5), we performed differential gene expression analysis using function "FindAllMarkers" and defined differentially expressed genes by passing the filtering criteria: discovery rate <0.01, log2(fold change)>0.5, mean log-transformed count in region >0.3. We then crossed the lists of temporal or regional differentially expressed genes to the *Imageable Genome* to identify imageable markers during early heart development.

### The Imageable Genome in human heart cells[21]
We downloaded single AnnData file containing the global raw counts from all source scRNA-seq via www.heartcellatlas.org. After removing cells annotated as "doublets" or "NotAssigned", we obtained 451,513 cells and 33,538 genes for downstream analysis. We used Scanpy toolkit 1.8.2 (Python v.3.8.2) to perform normalization

(normalize_per_cell: counts_per_cell_after: 10,000), log transformation (log1p) and differential gene expression analysis (rank_genes_groups: method = "wilcoxon").

We computed global cell type markers by pair each cell type versus the rest. We also computed regional cell type markers by pair in-region and out-region samples for each cell type. We kept the regions with more than 100 cells detected. We defined differentially expressed genes by passing the filtering criteria: adjusted $p$ value < 0.00001, log2(fold change) >5, and mean of log transformed count in group >0.5 for global cell type markers; adjusted $p$ value < 0.00001, a log2(fold change) >1, and a mean of log transformed count in group >0.3 for regional cell type markers. We then crossed the differentially expressed genes to the *Imageable Genome* to obtain either global cell type imageable markers or regional cell type imageable markers.

### The Imageable Genome in heart failure[27,28]

For the study by Koenig et al., we downloaded the complete differentially expressed gene lists analysed from 17 individuals with dilated (non-ischaemic) cardiomyopathy versus 28 control donors, and directly crossed to the *Imageable Genome*.

For the study by Wang et al., we downloaded scRNA-seq count matrix from the GEO database (accession codes: GSE109816 and GSE121893). After merging the data from the two repositories and removing cells annotated as "AV", we obtained a total of 11,056 cells and 25,742 genes derived from 4 hearts of patients with dilated cardiomyopathy (dHF), 2 hearts of patients with coronary heart disease (cHF) and 14 hearts of healthy donors. From "Seurat" (R package, version 4.0.5), we used function "NormalizeData" (normalization.method = "LogNormalize", scale.factor = 10,000) to normalized the raw counts, and function "FindMarkers" (test.use = "wilcox") to perform differential gene expression analysis. We computed the disease-cell type specific differentially expressed genes by pairing each heart failure group versus healthy group, or by paring healthy group versus merged two heart failure groups. We defined differentially expressed genes by passing the filtering criteria: an adjusted $p$ value < 0.01, |log2(fold change)| >0.25, and a mean of log transformed count in group >0.3. We then crossed the list of differentially expressed genes to the *Imageable Genome* to identify the imageable heart disease markers.

### The *Imageable Genome* in tumour development[31]

We downloaded RNA-seq count matrix of ovarian cancer patient no. 5 from the ovarian cancer study (10× platform, GEO Accession number: GSE158722). For downstream analysis, we used a total of 8037 epithelial cells from 3 collecting time points: Pre, sample collected before treatment; Start, sample collected right after treatment initiation; Relapse, sample collected at the first appearance of tumours biomarker MUC16 (CA-125) peak. From "Seurat" (R package, version 4.0.5), we computed gene expression changes in epithelial cells over time by using the function "FindMarkers" (test.use = "wilcox"). We defined differentially expressed genes by passing the filtering criteria: adjusted $p$ value < 0.01, |log2(fold change)| > 0.25, and the percentage of cells where the gene is detected in the case group >30%. We finally crossed the list of differentially expressed genes to *Imageable Genome* to identify imageable markers during ovarian cancer development.

### The *Imageable Genome* in metastasized tumours[32]

From the article's supplementary data, we downloaded differentially expressed gene list from pairwise comparisons between early-primary (tLung) versus advanced-stage primary (tL/B), or early-primary (tLung) versus metastatic (mLN, and mBrain) cancer cells. We performed additional differential gene expression analysis by comparing early-primary (tLung) to pleural effusion (PE) metastasis following the same methods and criteria described in this article. In brief, among the 396 malignant cells, we kept genes which expressed in >25% of cells within

either of the two compared groups. From "Seurat" (R package, version 4.0.5), we used function "FindMarkers" to pool out the differentially expressed genes. We determined the significance of the difference by two-sided Student's $t$ test with a Bonferroni correction. We defined differentially expressed genes by passing the filtering criteria: |log2(fold change)| > 0.585, two-sided Student's $t$ test $p$ value < 0.01, and adjusted $p$ value (Bonferroni) < 0.01. We then cross the list of genes to the *Imageable Genome* to obtain the imageable markers for lung cancer metastasis.

### The diagnostic *Imageable Genome* in human cancers

From the TCGA database, we downloaded clinical data and RNA-seq data (HTSeq−Counts and HTSeq−FPKM) of tumour or normal samples from TCGA database. We initially performed data cleaning by removing samples with duplicated aliquots, missing clinical information and removing cancer types with less than three normal solid tissue samples. After cleaning, 8620 samples from 21 cancer types were used for downstream analysis.

For each cancer type, we performed differential gene expression analysis between the normal tissues and tumour samples, by following the standard workflow as described in "edgeR" (R package, version 3.34.1) and "limma" (R package, version 3.48.3). In brief, we filtered out downregulated genes using the function "filterByExpr", calculated the between-sample (TMM) normalization factors by function "calcNormFactors" and "voom", we extracted the lists of differentially expressed genes by function "topTable", and cross them to the *Imageable Genome* to identify imageable markers for tumour detection.

For a given cancer type, we performed receiver operating characteristic analysis to determine the diagnostic value of a given imageable gene or reference gene. We computed the area under the curve (AUC) from the receiver operating characteristic curves by "pROC (R package, version 1.18.0) with function "roc".

### The prognostic *Imageable Genome* in human cancers

For a given cancer type, we used gene expression value to divide the patients into two subgroups. From "survminer" (R package, version 0.4.9), we determined the optimal cut-off for each gene expression by function "surv_cutpoint". To estimate the prognostic significance of each imageable gene, we initially computed univariable hazard ratios with 95% confidence intervals and corresponding $p$ values. We then performed multivariable survival analyses by fitting Cox proportional hazards models (CPH) that included patient age, tumour stage, gender and the expression of each imageable gene pooled from the univariable analyses at $p$ value < 0.05 (method section 18). For a given imageable gene, adjusted survival curves for CPH Models were generated and visualized using function "ggadjustedcurves" from R package "survminer", version 0.4.9. Then we performed log-rank tests to check the statistical significance of the survival difference between groups with high and low gene expression. We collected overall survival information from TCGA clinical data: "days_to_last_follow-up" if censored, or "days_to_death" if dead. We removed from the analysis any sample with vital_status = "alive" & overall_survival = 0. We did not consider "Stage" factor for Glioblastoma multiforme (TCGA-GBM) and Pheochromocytoma and Paraganglioma (TCGA-PCPG). We did not consider "Gender" for cervical squamous cell carcinoma and endocervical adenocarcinoma (TCGA-CESC), uterine corpus endometrial carcinoma (TCGA-UCEC) and prostate adenocarcinoma (TCGA-PRAD).

### The predictive *Imageable Genome* in human cancers[38]

We downloaded RNA-seq TPM matrix from the article's supplementary data. From ipilimumab-naive subset ($n = 84$), we made comparisons between responders (defined as having stable disease (SD), mixed response (MR), partial response (PR) or complete response (CR), $n = 47$) versus non-responders (defined as having progressive disease

(PD), $n = 37$), and evaluated the predictive value of each imageable gene by their area under the curve values derived from receiver operating characteristic analysis.

## The Imageable Genome in COVID-19[47,48]

For the study of Toni et al. (*NATURE* 2021), from their Supplementary Tables we extracted list of genes differentially expressed between 16 COVID-infected lungs versus 11 healthy lung and 18 COVID-infected hearts versus 21 healthy hearts across five major lung or heart cell types. For the study of Johannes et al. (Nature 2021), from their Supplementary Tables, we extracted the list of genes differentially expressed in 19 COVID-infected lungs versus 7 healthy lungs across four major lung cell types (AT1, AT2, monocyte and alveolar macrophage). Then we cross the two lists of differently expressed genes to the *Imageable Genome* to obtain the list of imageable COVID-19 lung/heart cell type related differently expressed genes.

## Reporting summary

Further information on research design is available in the Nature Portfolio Reporting Summary linked to this article.

## Data availability

The data generated in this study are provided in the Supplementary Information and Source data file. All raw data used in this study are publicly available and can be found as follows: Sc/snRNA-seq data from the human brain development study (Li, Science 2018) are available at PsychENCODE Knowledge Portal with Project SynID: syn4921369, under https://doi.org/10.7303/syn4921369. (SynapseID: syn17092080 and syn17092080). Count matrix and annotations have been deposited at http://psychencode.org. SnRNA-seq for Human brain development (Charles, Cell 2022) is available in the Gene Expression Omnibus (GEO) database under accession code: GSE168408. Bulk RNA-seq data from the ASD, SCZ and BP brain disease study (Gandal, Science 2018) are available at PsychENCODE Knowledge Portal with SynapseID: syn12080241 under https://doi.org/10.7303/syn12080241. SnRNA-seq data from the Alzheimer's disease study (Mathys, Nature 2019) are available at AD Knowledge Portal with SynapseID: syn18485175 under https://doi.org/10.7303/syn18485175. The respective contact for Synapse repositories is: Mette Peters, Director, Systems Biology Data Coordination Center, Mette@synapse.org. The raw sequencing data from human embryonic heart development study (Asp, Cell 2019) is under European Genome-phenome Archive (EGA) accession number: EGAS00001003996. The raw sequencing data from adult human heart cell atlas (Litviňuková, Nature 2020) are available at the European Nucleotide Archive (ENA) with accession number: ERP123138 (https://www.ebi.ac.uk/ena/browser/view/PRJEB39602). Count matrices and annotation are available for download from the Heart Cell Atlas (https://www.heartcellatlas.org). The respective contacts are the corresponding authors: J. G. Seidman (seidman@genetics.med.harvard.edu), Christine E. Seidman (cseidman@genetics.med.harvard.edu), Michela Noseda (m.noseda@imperial.ac.uk), Norbert Hubner (nhuebner@mdc-berlin.de), and Sarah A. Teichmann (st9@sanger.ac.uk). ScRNA-seq data from dilated cardiomyopathy (dHF) or coronary heart disease study (Wang, Nature Cell Biology 2020) are available in the GEO database under accession codes: GSE109816 and GSE121893. Sc/snRNA-seq from dilated cardiomyopathy (dHF) study (Koenig, Nature Cardiovascular Research, 2022) are available in the GEO database under accession code: GSE183852. Bulk RNA-seq data from human atrial fibrillation study (van Ouwerkerk, Nat Commun, 2019) are available in the GEO database under accession code: GSE127856. ScRNA-seq data from Ovarian cancer study (Nath, Nat Commun 2021) are available in the GEO database under the accession code: GSE158722. ScRNA-seq data from Lung cancer metastasis study (Kim, Nat Commun 2020) are available under GEO accession code: GSE131907. Sc/snRNA-Seq and bulk RNA-seq data from COVID-19 study (Toni, Nature 2021) are available in the GEO database under GEO accession code: GSE171668. ScRNA-seq data from COVID-19 study (Johannes, Nature 2021) are available in the GEO database under accession code: GSE171524, and in the single-cell portal: https://singlecell.broadinstitute.org/single_cell/study/SCP1219. Source data are provided with this paper.

## Code availability

The software code associated with the Imageable Genome is available on https://github.com/pablojane/ImageableGenome.

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

## Acknowledgements

The authors are grateful to Udo Schirp, Thomas Krause, Wolfgang Weber, Caius Radu and Johannes Czernin for their constructive and helpful discussion about the project. We also thank Andrew Tran for the design of Fig. 1. Cell-type icons in Fig. 3c and Fig. 4b–d, f, g, heart illustrations in Fig. 4a, b and biological illustrations in Fig. 4g were created using BioRender.com. M.A.W. acknowledges the support of the Swiss National Science Foundation (grant CRSII5-198569).

## Author contributions

P.J., X.X., V.T. and M.A.W. designed the project, and E.J., K.G., R.A.D., Y.G., F.K. and M.d.V.G. contributed to completing the design. P.J., E.J. and F.K. developed the AI pipeline structure, and P.J. and E.J. created the necessary software. X.X. developed the genomic pipeline, and created all resulting figures. P.J., X.X., V.T., E.J., K.G., R.A.D., Y.G., F.K., M.d.V.G. and M.A.W. acquired, analysed and interpreted pipeline data. P.J., X.X., V.T., R.A.D. and M.A.W. drafted the first manuscript, and E.J., K.G., Y.G., F.K., M.d.V.G. contributed to finalizing it. M.A.W. directed the project and guarantees the overall data integrity.

## Competing interests

The authors declare no competing interests.
