## [Peer Review File · Nature Communications]

The Imageable GenomeREVIEWER COMMENTS

Reviewer #1 (Remarks to the Author):

The authors have developed a pipeline to discover novel radiotracer-to-gene associations from text mining of pubmed articles. The approach uses available machine learning training framework and models from spacy. Using their pipeline, the authors report to have identified 9 times the number of NIH MICAD mentioned associations. The paper is clearly written but lacks some details on the methods which should be addressed. Specifically, lots of counts of genes are reported in the use cases and in the discussion section without any statistical relevance.

Comments:

Line 459: text categorizer model consisting of a convolutional neural network

From the text it is not clear what is used as input to the classifier? Was it full text or abstracts? Was any preprocessing performed on the train/test dataset?

Why was specifically this en_core_sci_md pipeline used?

Are convolution neural networks used because of the choice of Spacy as the framework? Can the authors provide an argument if CNNs are better than other machine learning algorithms for classifying texts?

Line 515: what is the prediction score 'with a prediction score above 0.5'. Is it comparable to an AUC score? Why is the threshold set at 0.5?

Line 101: what is the sequence and structure similarity of the imageable genes (n=1173) identified by the authors? How many of these were previously known?

For improving explainability of the predictions, can the text mining approaches also highlight or select the portion of text that is associated with the radiotracer-gene relationship?

Cell-type specificity using single-cell transcriptomics data:

The single-cell dataset has only 762 cells. This is a rather low number for cell-type specificity analyses. The method described for determining marker genes - findallmarkers in Seurat does not take into account specificity of a gene for a given cell-type.

Line 145: how can this method be used to quantify cellular composition?

Line 146: authors should provide a disease vs normal (and early vs late) comparison in a cell-type specific manner to identify imageable genes in a disease state for both brain and Alzheimer's dataset.

For the patient level view of their - are the genes identified highly expressed at the patient level in the cell-types of interest? In general, for both single-cell and bulk datasets used in this study, it would be good to know the patient-level applicability of the identified genes.

Disgenet disease association - how is the disease association determined? Is it based on the DSI or DPI score? Are the genes uniquely associated with a certain class of diseases?

Application in cardiology and oncology - here, as with the other use cases presented before, it would be good to know the specificity of the genes identified, and then also show the applicability at the patient level by showing per-patient pseudo-bulked gene expression of selected genes.

Line 254: 47 imageable genes are differently - Is that statistically significant?

Line 287: The Imageable Genome amounts to 1.8% of the human genome

Does this take into account the Imageable genome for which tracers have not yet been developed?

Would it be in the scope of this study for the authors to define what are the specific properties / rules of the Imageable genome, for example, some interpretable features that could be used to assess yet un-implicated genes?

Line 292: 'most likely affect the expression' : 'The implication that disease or development or severity of disease likely impacts genes that are part of the imageable genome needs to be backed up with statistical evidence. How many genes are implicated in various diseases? How many of those are imageable? Are they implicated in disease based on just differential expression analysis or is there any causal link?

Reviewer #2 (Remarks to the Author):

This is a commendable large-scale effort on data mining, which used AI/NLP pipelines to link individual genes with their relevance in human disease, and with specific molecular imaging tracers. Data from over 55,000 individuals, including patients and healthy controls, were used in this data mining effort, and tens of millions of database entries were searched and matched. The study resulted in a list of over 1,000 imageable genes, which have clinical relevance and at the same time are potential targets for molecular imaging tracers. The paper discusses these findings categorized from the perspective of applications to Neurology, Cardiology, and Oncology. Overall, this is a very significant study of broad interest and potential impact, as it can inform numerous studies using molecular imaging end points in drug discovery and development.

Some individual points:

-- Although lots of data was mined in order to arrive at a number of SNP targets, tissue data from only 3 adult brains (and 7 cardiac specimens, for the cardiologic findings) were used to test the differential gene expression related to 17,093 SNPs of interest. It is unclear whether such a small number of brains can capture any reasonable variability, however it is recognized that a larger scale study on analyzing gene expression from brain tissue is very demanding and potentially beyond the scope of the current paper. In general, I am somewhat confused about how many samples were used for analysis of gene expression, because there seems to be conflicting information throughout respective sections.

--The AI literature has seen lots of misleading (incorrectly interpreted) and non-replicable results, largely due to massive fishing expeditions that are not properly cross-validated or replicated on independent samples (this reviewer is an AI researcher). I would want to see evidence that the findings of this data mining analysis were quality controlled, checked, and ideally replicated (this is admittedly very difficult for a study of this magnitude, so I welcome thoughts of the authors on this issue). Although the tissue-based experiments would presumably alleviate this concern to some extent, they are very small, compared to the magnitude of data mining that was performed. At least some basic testing of the findings using interpretable metrics should be presented.

--It is not clear what that this argument in the Introduction is valid: "One reason for this translational bottleneck is linked to the fundamental lack of knowledge concerning the entirety of molecules that can be targeted with the repertoire of available molecular imaging agents". These radiotracers that don't make it to the clinic have been developed and their use has been demonstrated---cost, availability, and other factors are likely to limit clinical adoption. In general, the paper tends to over-claim, at times (for example, "The Imageable

Genome can serve as a 'Rosetta stone' that systematically translates these complex

genomic discoveries into easily applicable clinical imaging tests” claims to address the miniscule rates of translation of genomic findings into FDA-regulated tests. However it is not clear whether and how this primarily data mining study will change this figure).

--Much of the value of the current study relies on the availability of an easily accessible and interpretable software or web implementation to disseminate the findings. This is not discussed in the paper.

--The naming of the tracers should probably be checked carefully. I spotted one typo related to a tracer I am personally familiar with: it is [18F]NOS, and not [18F]NOS-9

Reviewer #3 (Remarks to the Author):

The manuscript entitled, “The Imageable Genome,” details the process by which the authors used text classifiers to identify an updated set of PET-compatible imaging tracers/targets, including many studies, associations, and targets not in the current NIH-supported Molecular Imaging and Contrast Agent Database (MICAD). The authors go on to detail three case uses for The Imageable Genome dataset, specifically in the brain, heart, and cancer, and identify several novel uses for existing tracers. The work presented here is very exciting and is of high interest for its readily translatable clinical and basic science applications.

Below are some comments and questions related to the work presented in no particular order:

1. This work is very exciting, and it seems like it would be a shame to have the identifications buried in supplemental tables. It seems like this work could better serve the community as part of a standalone database (or an update to NIH MICAD) that could be updated as more is learned. The authors should strongly consider doing this.
2. The authors chose to focus on neurology, cardiology, and oncology for the case uses as they identified many imageable genes already associated with these diseases (Fig. 2C, lines 113-121). As I understand it, the identification may be self-fulfilling as many of these tracers were developed for those purposes; however, the stated goal of “The Imageable Genome” is to expand these molecular imaging tools into new areas. Could the authors apply the same methodology to an organ and disorder not widely diagnosed using PET or PET-CT currently to demonstrate the untapped potential to community?
3. On lines 169-170 (and it comes up again in the cardiology and oncology sections), the authors state, “Finally, while the Imageable Genome suggests promising imaging targets for already existing

radiotracers, it also guides the development of novel radiotracers.” How exactly does this study guide the development of novel radiotracers? It seems to me that this is just a wish list of genes/proteins that it would be great to have a radiotracer that could be used to detect.

4. The authors often only identify how many differentially expressed genes are imageable. For example, as stated only 4% and 9% of differentially expressed genes in the embryo/adult heart stages are imageable. Would it be possible for the authors to compile a list of imageable genes (irrespective of differential expression) across different organs based on expression data from databases like GTEX, Human Protein Atlas, or similar?

5. Minor critique, I noticed that the text does not refer to all the panels of Figure 4 and that some of the panels are misattributed. Specifically, “we identified 6 imageable genes that are predictive for the onset of atrial fibrillation (Fig. 4d)”; however, this appears to Fig. 4E in the figure.

Reviewer #1

The authors have developed a pipeline to discover novel radiotracer-to-gene associations from text mining of pubmed articles. The approach uses available machine learning training framework and models from spacy. Using their pipeline, the authors report to have identified 9 times the number of NIH MICAD mentioned associations. The paper is clearly written but lacks some details on the methods which should be addressed. Specifically, lots of counts of genes are reported in the use cases and in the discussion section without any statistical relevance.

Comments:

1.) Line 459: text categorizer model consisting of a convolutional neural network
From the text it is not clear what is used as input to the classifier? Was it full text or abstracts? Was any preprocessing performed on the train/test dataset?

Reply: We are grateful for the opportunity to clarify these two important points. Firstly, we used abstracts texts as input to the classifier for both, training and testing. We specified this point in the methods part of the manuscript as follows:

“We used these 2’997 PMIDs to search for the corresponding PubMed citations in the parquet files, and obtained a list of 2’060 abstracts containing a radiotracer for molecular imaging. To these, we added 2’308 random abstracts from PubMed not containing radiotracers. We verified the correct labelling of all abstracts by a team of experts in the field using the collaborative annotation platform Doccano ⁷⁰, and generated a training corpus of 4’368 annotated abstract texts. Prior to the model training, we performed pre-processing by removing any leading and trailing spaces, making all the characters lowercase and eliminating all punctuation signs excepting brackets, dashes and percentage symbols. We then performed a 75/25% train/test split of this training corpus and trained a text categorizer model consisting of a convolutional neural network using python’s natural language processing tool spaCy ⁷¹, and the pre-trained model “en_core_sci_md” from ScispaCy ⁷², chosen for its efficiency, its state-of-the-art performance, and its compatibility with spaCy.”

(Page 30)

Secondly, we clarified the preprocessing step where we removed any leading and trailing spaces, converted all characters to lowercase and eliminated all punctuation signs except brackets, dashes and percentage symbols. Initial tokenization of the input texts and vectorization of the resulting tokens are a part of the pipeline of the spaCy model and therefore, have been considered layers of the model rather than part of pre-processing stage. We added this information to the manuscript as it follows:

“Prior to the model training, we performed pre-processing by removing any leading and trailing spaces, making all the characters lowercase and eliminating all punctuation signs excepting brackets, dashes and percentage symbols.”

(Page 30)

2.) Why was specifically this en_core_sci_md pipeline used?

Reply: Our choice of the scispaCy en_core_sci_md model followed the same reasoning as our choice of the spaCy framework for all the Natural Language Processing tasks in the *Imageable Genome* data pipeline: We were looking for a python-based, robust, efficient, state-of-the-art model that was trained on a large biomedical text corpus, and therefore could capture the specific vocabulary and contextual dependencies that could help differentiate two different categories of pubmed abstracts. As described by Neumann et al. (*Proceedings of the 18th BioNLP Workshop and Shared Task*. 2019), both en_core_sci_sm and en_core_sci_md models show similar processing speeds to the fastest C++ and Java models, specifically developed for production purposes, and similar accuracy for the tasks of Part of Speech Tagging and Dependency Parsing. Regarding the size of the model which suited best our project, we chose the medium size model (en_core_sci_md) as it utilizes a larger vocabulary (en_core_sci_sm: 58'338, en_core_sci_md: 101'678) while keeping a similar performance speed and benefiting from the use of word vectors that could help obtain a more accurate representation of the input text. We included our reasoning in the methods part as follows:

“We then performed a 75/25% train/test split of this training corpus and trained a text categorizer model consisting of a convolutional neural network using python’s natural language processing tool spaCy⁷¹, and the pre-trained model “en_core_sci_md” from ScispaCy⁷², which we chose for its efficiency, state-of-the-art performance, and compatibility with spaCy.”

(Page 31)

3.) Are convolution neural networks used because of the choice of Spacy as the framework? Can the authors provide an argument if CNNs are better than other machine learning algorithms for classifying texts?

Reply: CNNs have been used in the development of the text classifier largely due to the choice of spaCy as a framework, which offers pre-built architectures that include convolutional layers. The use of convolution operations allows the text classifying model to learn information on words relative to their context (Kim et al. *Proceedings of the 2014 Conference on Empirical Methods in Natural Language Processing*. 2014; Zhang et al. *Character-level Convolutional Networks for Text Classification*. 2015; Conneau et al. *Very Deep Convolutional Networks for Text Classification*. 2017), ultimately being able to identify whether certain clusters of words contain relevant information regarding the categories to which the text belongs or not after proper training. Within the spaCy v2 framework, CNN-based models represented the best option since, according to the official software documentation (<https://v2.spacy.io/api/textcategorizer>), the alternative text classifying architectures available consisting of simple bag of words (BoW) models may not be as accurate, especially when texts are short, as it is the case for abstract texts.

4.) Line 515: what is the prediction score ‘with a prediction score above 0.5’. Is it comparable to an AUC score? Why is the threshold set at 0.5?

Reply: Thank you for the opportunity to clarify this point. We trained the text classifier model using two exclusive categories, which we refer to as “positive” and “negative”, determining whether the abstract texts do or do not contain a radiotracer for molecular imaging, respectively. Given the exclusivity of these categories, the output layer of the model consists of a softmax function that assigns numerical values between 0 and 1 to each category, all of which add up to exactly 1. For a given input abstract text, the model output value assigned to each category can then be interpreted as a prediction of the probability of it belonging to that category. Hence, the “prediction score” of the text classifier model, in this work, refers to the numerical value assigned by the text classifier to the “positive” category. The model prediction score threshold used to classify an abstract text as “positive” has been set to 0.5 since the main goal at this stage of the pipeline is to maximize the recall score of the model by minimizing the number of false negative classifications. Thus, the presence of false positive classifications (lower classifying precision score) has not been considered as critical for the proposed application and using a higher prediction score threshold has been deemed too restrictive. The training scores of the model given in lines 463 and 464 of the original manuscript correspond to the set prediction score threshold of 0.5, having a mean recall value of 98.8% over 10 complete training processes. We clarified this aspect and the fact that the given precision, recall and F1 score are averages of 10 different training processes over 10 iterations in the revised manuscript:

“We trained the model 10 times over 10 iterations to classify abstract texts describing a radiotracer with an average precision of 97.8 %, an average recall of 98.8 %, and an average F1 score of 98.3%, with a set prediction score threshold of 0.5, being the numerical value given by the softmax function in the output layer of the model to the “positive” category.”

(Page 31)

We set the threshold value to 0.5, following a rule of thumb as it has an equal distance to 0, the absolute negative prediction score and to 1, the absolute positive prediction score. Based on the reviewer’s question, we present the respective AUC analysis revealing decent precision and recall for this value as shown in **Extended Data Fig. 1**:

Extended Data Fig. 1 ROC curve obtained after 10 complete training processes. The AUC value is presented as Mean ± Standard Deviation. Each individual ROC curve has been built using a prediction score threshold resolution of 0.02 (from 0 to 1 in increments of 0.02) and their corresponding AUCs have been computed using numerical integration (trapezoid method) and validated using the exact methods proposed in Algorithms 1 and 2 of reference “Fawcett, Tom. (2004). ROC Graphs: Notes and Practical Considerations for Researchers. Machine Learning. 31. 1-38.”

5.) Line 101: what is the sequence and structure similarity of the imageable genes (n=1173) identified by the authors? How many of these were previously known?

Reply: We appreciate the reviewer’s question, which we had also posed ourselves during the completion of the project, as a common sequence or structure within the *Imageable Genome* might allow predicting the next genes that will be imageable in the future.

In order to systematically identify common sequence or structure, we performed a systematic homology comparison using the NIH Basic Local Alignment Search Tool (BLAST, <https://blast.ncbi.nlm.nih.gov/Blast.cgi>), which compares nucleotide or protein sequences to sequence databases and computes the statistical significance to find regions of similarity between biological sequences:

- First, we performed a comparison with the most reliable search tool BLASTP (protein-protein BLAST), which did not identify a single common sequence.
- Second, we performed a comparison with maximal sensitivity using PSI-BLAST (Position-Specific Iterated BLAST), which did not identify a credible common sequence. The only sequence identified was that of the protein aggrecan, which is not even part of the *Imageable Genome*.
- Third, we performed a comparison using DELTA-BLAST (Domain Enhanced Lookup Time Accelerated BLAST), which is also set to maximal sensitivity. This engine identified only the mucin family of proteins, of which 17 members are part of the *Imageable Genome*. This result most likely reflects the enrichment of the *Imageable Genome* and the database in extracellular matrix proteins such as mucins.

Overall, the *Imageable Genome* comprises a wide variety of the different protein classes, as presented below using the protein classification software PANTHER (pantherdb.org). This wide variety is probably the reason for the finding that there is no common sequence and structure similarity within the *Imageable Genome*.

However, there are functional similarities within the *Imageable Genome*. For example, a pathway analysis using pathview (<https://pathview.uncc.edu/>) reveals that the vast majority of human neuroactive receptors are currently imageable. It is anticipated that all human neuroactive receptors will be imageable in the future.

6.) For improving explainability of the predictions, can the text mining approaches also highlight or select the portion of text that is associated with the radiotracer-gene relationship?

Reply: The current project is based on a human-AI-hybrid pipeline, in which humans took over exactly the function described by the reviewer. Thus, our AI was not developed to highlight or select the portion of text that is associated with the radiotracer-gene relationship. It was developed to search for the presence or co-occurrence of entities of the types of protein or gene and radiotracer within the abstract or the title text. However, we are currently working on a new AI tool utilizing a fine-tuned Large Language Model to independently detect these types of associations.

7.) Cell-type specificity using single-cell transcriptomics data: The single-cell dataset has only 762 cells. This is a rather low number for cell-type specificity analyses.

Reply: The reviewer raises a valid point. Cell number and sequencing depth are two key factors for every single-cell RNA-seq project. We agree that 762 cells is a rather low number for cell-type specificity analyses; yet, we believe that this number is sufficient for the selected analyses for the following reasons:

- First, a low cell number can be compensated by deep sequencing and sample size (Zhang, M.J., *Nature Communications* 2020). Li et al (*Science*, 2018) established the used data set from 8 foetal brains, and compensated for the relatively low cell number with a high sequencing depth of about 1.5 million uniquely mapped reads per cell. The in-depth sequencing allowed enough interrogation of human transcriptome at single cell level (60'155 genes included with at least 1'000 genes having RPKM > 1 per cell).
- Second, the authors used iteratively a clustering and classification approach (Lake, 2016 *Science*), labelled cell clusters on the basis of well-known cell type markers. These methods are essential to identify cell clusters and annotate cell types/subtypes based on well-known markers for the scRNAseq database. This is a manual step that can also compensate the low number of cells used in the study. Also, they validated the data robustness by a second clustering approach from Seurat package, to ensure a reliable cell type annotation.
- Third, the cell number was sufficient to generate significant results. We used the Seurat function FindAllMarkers to identify marker genes for each individual cell type, with strict criteria: For any given comparison, we only considered genes that were expressed by at least 30% of cells in either population. A difference between percent of cells expressing gene within and outside group > 30%, and a log2 fold change > 1. Genes that exhibit a false discovery rate under 0.00001 were considered statistically significant. These criteria guarantee the specificity of the cell type marker analysis. The expression of top 5 cell type markers exhibiting the highest specificity score for the prenatal scRNA-Seq database are also shown by heatmap in the original paper's Fig. S22.
- Fourth, we likewise were able to obtain 61 significant cell type markers for 6 foetal brain cell types (**Extended Data Fig. 4**).

However, we completely agree with the reviewer that the availability of more cells might allow to identify even more markers, also with a higher certainty. Based on the reviewer's comment, we mentioned this limitation in the discussion section of the manuscript:

"The availability of more single cell sequencing data in the future might allow to identify even more imageable genes with relevance in human diseases."

(Page 15)

8.) The method described for determining marker genes - findallmarkers in seurat does not take into account specificity of a gene for a given cell-type.

Reply: We apologize for the potential misunderstanding. Findallmarkers in Seurat actually does take into account specificity of a gene for a given cell-type, by defining thresholds under the FindAllMarkers() function.

For our analyses, we specified thresholds for the minimum percentage of cells expressing the gene in either of the two types of cells using the “min.pct” subfunction with at least 30% cells in a tested cell type and less than 30% cells in the rest cell types, “pct_diff” difference between percent of cells expressing gene within and outside a tested cell type > 30%, a false discovery rate < 0.00001, and a log2 fold change > 1.

We clarified this point in the manuscript as follows:

“To identify marker genes with a specificity for a given cell type, we compared each group of cells to the rest of the cells and defined cell type specific markers by passing the filtering criteria: for each comparison, the percentage of cells where the gene is detected in the group > 30%, the percentage of cells where the gene is detected outside the group < 30%, a difference between percent of cells expressing gene within and outside group > 30%, a false discovery rate < 0.00001, and a log2 fold change > 1.”

(Page 38)

9.) Line 145: how can this method be used to quantify cellular composition?

Reply: Thank you for the opportunity to clarify this point. In general, a cell-binding radiotracer is injected, and imaging is performed to assess the concentration of the injected radiotracer in a specific tissue, which then correlates with the cell number in this tissue. Examples include:

- Beckford-Vera et al. *NATURE Communications*. 2022: The HIV neutralizing antibody 89Zr-VRC01 was injected in HIV-infected individuals and non-infected controls. PET imaging was performed at different time points, and the relationship between PET tracer uptake in tissues and number of CD4 T cells in lymph node was verified via flow cytometry in biopsies.
- Woodham et al., *NATURE Methods*. 2020: PET with radiolabelled dimeric major histocompatibility molecule scaffolds was used to quantify the tissue concentration of antigen-specific CD8+ T-cells. The results were verified by ELISpot.
- Ly et al., *NATURE Clinical Practice*. 2008: The paper summarizes PET-based techniques to quantify the tissue concentration of stem progenitor cells in the myocard.
- Seo et al., *NATURE Communications*. 2020: PET with radiolabeled capsid from adeno-associated viruses that bind the LY6A receptor was used to quantify brain endothelial cells. A multichelator with fluorescence was used to confirm the imaging results.

10.) Line 146: authors should provide a disease vs normal (and early vs late) comparison in a cell-type specific manner to identify imageable genes in a disease state for both brain and Alzheimer's dataset.

Reply: We hope that we understood the question correctly, as we believe that we did exactly what the reviewer requested. We utilized data from the paper of Mathys et al. (*NATURE*, 2019) of 15 individuals with early Alzheimer's, 9 individuals with late Alzheimer's, and 24 healthy controls. Using this dataset, we performed two comparisons:

- We compared early Alzheimer's vs healthy brains, and listed all cell type specific imageable genes for early Alzheimer's in **figure 3d, 3e, and supplementary table 8**.
- Then, we compared late Alzheimer's vs early Alzheimer's, and listed the cell type specific imageable genes for late Alzheimer's in **supplementary table 8**.

We clarified this point in the manuscript as follows:

*"Analysing 80'660 single-nucleus transcriptomes from the prefrontal cortex of individuals with varying degrees of Alzheimer's disease, we identified 41 cell-type specific imageable genes up or down-regulated exclusively in AD-early diseased versus Normal brains (**Fig.3d, Supplementary Table 8**), and 81 cell-type specific imageable gene up or down-regulated exclusively in AD-late diseased versus Ad-early diseased brains (**Supplementary Table 8**)."*

(Page 8)

11.) For the patient level view of their - are the genes identified highly expressed at the patient level in the cell-types of interest? In general, for both single-cell and bulk datasets used in this study, it would be good to know the patient-level applicability of the identified genes.

Reply: To answer the reviewer's question, we performed patient-based analyses for all single-cell and bulk datasets used in our study: First, we analysed the expression for each cell type specific imageable gene that is diagnostic for early Alzheimer's at patient level, and demonstrate the results in the new **Extended Data Fig. 5a**. Then, we performed the same analysis using the bulk RNAseq data for autism spectrum disorder, bipolar disorder, and schizophrenia (**Extended Data Fig. 5b**). Last, we repeated the analysis for haptoglobin expression across 6 major heart cell types in dilated heart failure, coronary heart failure or normal hearts at patient level (**Extended Data Fig. 5c**).

Overall, we found that 27% to 90% of significantly overexpressed imageable genes identified in single-cell and bulk datasets remained significantly overexpressed at patient level. We present these overall results in **Supplementary Table 8 and 9**, referenced on page 8 of the manuscript.

Extended Data Fig.5 Elevated expressions of cell type specific Alzheimer's disease Imageable Genome genes at patient level. Imageable Genome genes with significantly elevated expression in **a**, AD-early patients (n=15) versus normal donors (n= 24) or AD-Late (n=9) versus AD-early patients and **b**, in ASD (n=43) /SCZ (n=558) /BP (n=216) patients versus normal heart donors (n=986). **c**, Haptoglobin expression in each cell type from dilated heart disease (n=4) or coronary heart disease (n=2) versus normal condition (n=14). Dot: individual patient. P-values by two-sided Wilcoxon rank-sum test are shown by asterisk. *p<0.5, **p<0.01, ***p<0.001, ****p<0.0001, ns: no significance.

12.) Disgenet disease association - how is the disease association determined? Is it based on the DSI or DPI score? Are the genes uniquely associated with a certain class of diseases?

Reply: We determined the disease association using the GDA (Gene-disease association) Score. We did not use the Disease Specificity Index (DSI), which evaluates the specificity of a gene associated to a number of diseases. We also did not use the Disease Pleiotropy Index (DPI), which evaluates the similarity among the diseases a gene associated to.

To extend the aspects of the associations, we analysed the distribution of the DSI and DPI score and the disease associations for 1173 imageable genes. From the distribution of the two scores, we see that the majority of imageable genes are not uniquely associated with a certain class of disease. We have prepared the results as **Extended Data Fig. 2**, as a complementary source for **Figure 5c(i)**, referenced on page 6 and page 34 of the manuscript.

Extended Data Fig. 2 Disease severity index (DSI) and Disease Pleiotropy Index (DPI) distributions of Imageable genome genes. Histograms showing the distribution of **a.** DSI Scores and **b.** DPI scores for 916 disease-associated imageable genes. **c.** A scatter plot of DSI and DPI, with top values are shown for DSI (DSI score >0.9, 7 genes) and DPI (DPI score >0.95, 11 genes), and **d.** a legend to show their uniquely (or high similarly) associated disease class.

13.) Application in cardiology and oncology - here, as with the other use cases presented before, it would be good to know the specificity of the genes identified, and then also show the applicability at the patient level by showing per-patient pseudo-bulked gene expression of selected genes.

Reply: Based on the reviewer's comment, we performed patient-based analyses in cardiology and oncology: First, we re-analysed the expression of the cell type specific marker haptoglobin in heart failure vs the healthy heart at patient level (**Extended Data Fig. 5b**). Then, with the top 10 up-regulated diagnostic imageable genes for the 5 cancer types presented in **Fig. 5c**, we present the patient-level results using scatter plots in the new **Extended Data Fig. 7**. Each dot represents an individual patient.

We reference the new extended data figure on page 12 of the manuscript.

Extended Data Fig. 7 Expression of top 10 diagnostic *Imageable Genome* genes across 5 TCGA cancer types at patient level. P-values by two-sided Wilcoxon rank-sum test are shown by asterisk. ****p<0.0001.

14.) Line 254: 47 imageable genes are differently expressed - Is that statistically significant?

Reply: To answer the reviewer's question we assessed statistical significance for the AUCs of the 47 predictive imageable genes using the R function from package verification roc.area() (version 1.42). This function calculates the area underneath a ROC curve following the process outlined by Mason, S.J. and Graham, N.E (Q.J.R. Meteorol. Soc., 2002). The p-value is calculated using the Wilcoxon rank-sum test. The p-value addresses the null hypothesis H_0 : The area under the ROC curve is 0.5 i.e. the forecast has no skill. We calculated the p values by two directions: test > control or control > test, depending on the AUC values. Overall, we found statistical significance (p value <0.05) in 34 of 47 genes:

Gene	AUC	P
OVGP1	0.66444937	0.00787827
MMP9	0.65742794	0.01042035
FOLR2	0.64745011	0.0152247
CNR2	0.61382114	0.01943512
CD22	0.63747228	0.0218086
ITGA6	0.63562454	0.0230692
KRT9	0.60716925	0.02592058
IL1R2	0.6286031	0.02956519
MGLL	0.62601626	0.03213415
ITGAX	0.62453806	0.03375723
ITGAM	0.61899483	0.0404596
CD163L1	0.61640798	0.04387813
ABCC3	0.61419069	0.04699366
DPP4	0.61419069	0.04699366
SLC16A13	0.61271249	0.04916001
PCYT1B	0.29194383	0.00099675
GRIK2	0.31707317	0.00360799
TGFB2	0.31892092	0.00391838
GRIA4	0.32224686	0.00438643
GSK3A	0.32298596	0.00439183
ADRB1	0.32815965	0.00462948
RYR3	0.32815965	0.00580237
NDUFA11	0.32963784	0.00588641
GRPR	0.32889874	0.00597557
HTR3E	0.3481153	0.00606453
TF	0.33407243	0.00740671
MC5R	0.3466371	0.00805621
MMP10	0.3481153	0.00920178
CHRM5	0.34146341	0.00960727
EMCN	0.34405026	0.01103209
ADORA1	0.345898	0.0115631
EPCAM	0.3481153	0.01269835
NECAB2	0.34922395	0.01270569
GYS1	0.3488544	0.01295399

We included these results into **Supplementary Table 18** and updated the manuscript as follows:

“For example, 47 imageable genes are differently expressed in melanomas sensitive or resistant to PD1-blockade ³⁷ (in 34 cases with a p < 0.05, Fig 5d, Supplementary Table 18), including the matrix metalloproteinase MMP9, the adenosine receptor ADORA1, the glycogen synthase kinase GSK3A, the folate receptor FOLR2 and the transforming growth factor TGFB2 (Fig. 5d).”

(Page 13)

15.) Line 287: The Imageable Genome amounts to 1.8% of the human genome Does this take into account the Imageable genome for which tracers have not yet been developed?

Reply: The *Imageable Genome* only refers to the genes whose expression is imageable today. Of course, it is possible, yet unlikely, that the expression of the entire human genome will be imageable one day.

While it remains difficult to predict for which gene products there will be tracers in the future (see answer to question 5), the *Imageable Genome* however does indicate the gene products for which tracers would be highly desirable.

16.) Would it be in the scope of this study for the authors to define what are the specific properties / rules of the Imageable genome, for example, some interpretable features that could be used to assess yet un-implicated genes?

Reply: As all co-authors have a background in radiotracer development, we are highly interested for using the *Imageable Genome* to direct future radiotracer development. From our experience we can say that gene products that are desirable to target in the future should have a clinically-relevant and validated diagnostic, prognostic and/or predictive meaning, should be highly expressed in comparison to the surrounding tissues, and should be easily accessible, which implies most of them will be transmembrane or extracellular. The *Imageable Genome* data did not change this paradigm.

It has to be noted though that the *Imageable Genome* does not represent the sum of the most desirable genes to image. In contrast, we found that for most diseases the most relevant diagnostic, prognostic and/or predictive genes are not yet imageable. We believe that our project will allow making the most of the imaging possibilities that we have, but in the best case it should induce a change in radiotracer development to focus on the most clinically-relevant targets.

17.) Line 292: ‘most likely affect the expression’: ‘The implication that disease or development or severity of disease likely impacts genes that are part of the imageable genome needs to be backed up with statistical evidence. How many genes are implicated in various diseases? How many of those are imageable? Are they implicated in disease based on just differential expression analysis or is there any causal link?’

Reply: To answer this important question we constructed **Extended Data Fig. 11**, in which imageable genes are pooled by differential expression analysis between diseased and normal conditions. We see roughly 1% to 30% of the human protein coding genome identified as disease related markers across brain, heart and cancer fields, among which 21% are imageable genes.

Extended Data Fig. 11 Composition of *Imageable Genome* genes/*non-Imageable Genome* genes in the complete T2T-CHM13 human genome assembly. Donut plots representing the number of unique imageable markers (red) and unique non-imageable markers (yellow) implicated in various diseases. The green part in each donut is the remaining total number of protein coding genes.

We included this part into the manuscript as follows:

“So far, about 1 to 30% of human protein coding genes have been identified as disease related markers in neurology, cardiology and oncology, among which 21% are imageable genes (Extended Data Fig. 11).”

(Page 15)

Moreover, some of the Imageable candidate genes are validated in other publications that to be causal in the related disease. For each disease type, we list one example candidate with causal link to the related disease:

- Alzheimer's disease (AD): Glial fibrillary acidic protein (GFAP), is an astrocytic cytoskeletal protein that has been shown to be a marker of abnormal activation and proliferation of astrocytes and its elevated expression has shown to be associated to individuals with early-onset AD.
- Autism Spectrum Disorders (ASD): The urokinase plasminogen activator receptor (PLAUR) is reported to be a risk gene for ASD by modulating MET signalling.
- Schizophrenia (SCZ): CX3C chemokine receptor 1 (CX3CR1), is a seven-transmembrane domain Gi protein-coupled receptor expressed by microglia bound by ligand CX3CL1. CX3CL1-CX3CR1 activation and CX3CR1 polymorphisms have been reported to be causal for SCZ progression. CX3CR1 expression is decreased in SCZ reported in various studies.
- Dilated heart failure (DCM): Creatine kinase (CK), is a major phosphotransfer system in the heart energy production to energy utilisation and an important serum marker for myocardial infarction. Down-regulation of the CK system is a hallmark of heart failure.
Ovarian cancer: We identified CD74 as an imageable marker elevated in late and relapse stage as compared to pre-treatment in ovarian cancer. CD74 is a type II transmembrane protein. Its expression level is reported to be associated to ovarian tumour grade and promote cervical carcinogenesis via oncogenic signalling mechanisms and may serve as a potential antitumour target.
Folate receptor alpha (FOLR1), encodes a glycosylphosphatidylinositol (GPI)-anchored cell-surface glycoprotein (FR α), is reported to be associated to ovarian tumor disease burden and treatment outcomes.
- Cancer diagnostic and/or prognostic genes: Heat shock protein 90 alpha family class A member 1 (HSP90AA1), has an elevated expression across more than 20 cancer types, and has been reported to be a prognostic marker in 6 cancer types (Liver Hepatocellular Carcinoma, Kidney renal clear cell carcinoma, head and neck squamous cell carcinomas, Lung adenocarcinoma, Breast cancer and Mesothelioma).

Reviewer #2

This is a commendable large-scale effort on data mining, which used AI/NLP pipelines to link individual genes with their relevance in human disease, and with specific molecular imaging tracers. Data from over 55,000 individuals, including patients and healthy controls, were used in this data mining effort, and tens of millions of database entries were searched and matched. The study resulted in a list of over 1,000 imageable genes, which have clinical relevance and at the same time are potential targets for molecular imaging tracers. The paper discusses these findings categorized from the perspective of applications to Neurology, Cardiology, and Oncology. Overall, this is a very significant study of broad interest and potential impact, as it can inform numerous studies using molecular imaging end points in drug discovery and development.

Some individual points:

1.) Although lots of data was mined in order to arrive at a number of SNP targets, tissue data from only 3 adult brains (and 7 cardiac specimens, for the cardiologic findings) were used to test the differential gene expression related to 17,093 SNPs of interest. It is unclear whether such a small number of brains can capture any reasonable variability, however it is recognized that a larger scale study on analyzing gene expression from brain tissue is very demanding and potentially beyond the scope of the current paper. In general, I am somewhat confused about how many samples were used for analysis of gene expression, because there seems to be conflicting information throughout respective sections.

Reply: Thank you for raising this important point. In Figure 1, we summarized the total samples used for major conditions of each study. It is true that within each study, there are several detailed conditions using part of the samples for specific analysis (for example, in Brain development the author used in total 41 brains for bulk RNA-seq data collection, but 3 adult brains out of 41 brains to collect snRNA-seq data, and 8 fetal brains out of 41 brains to collect scRNA-seq data). We have not summarized the detailed condition within each study, but described them individually in the method part.

Based on the reviewer's question, we introduced the recently published paper in Cell 2022, "Human prefrontal cortex gene regulatory dynamics from gestation to adulthood at single-cell resolution", by Charles A. Herring, as a validation dataset for the Li, 2018 *Science* paper. In Charles's paper, they collected and analysed snRNA-seq data of 26 post-mortem prefrontal cortex (PFC) samples from individuals spanning foetal, neonatal, infancy, childhood, adolescence, and adult stages of development. One of their major findings is that they identified 14,984 unique development-associated differentially expressed genes (devDEGs, false discovery rate [FDR] <0.05) within at least one major trajectory (cell type). The author has also confirmed the similarity of the devDEGs dynamics between their data and Li, 2018 bulk RNA-seq dataset. (Described in Figure S3F, Charles A, 2022)

We used the devDEGs table to cross to Li's bulk RNAseq data, and constructed an **Extended Data Fig. 3** to show the number of overlapped imageable genes, and top 5 up-regulated or down-regulated foetal specific imageable genes present in both papers. This figure is referenced on page 7 of the manuscript.

Extended Data Fig. 3 The validation set for human brain development (Charles, Cell 2022). Venn diagrams to show the number of overlapped imageable development markers identified from Li, 2018 study (bulk RNAseq data, yellow circle) and Charles, 2022 study (snRNAseq data, green circle). For each cell type, the expression patterns of top 10 genes (top 3 genes for Mature Oligodendrocytes) that commonly upregulated in fetal stage for both studies are shown by scatter plots. The expression values are from Li, 2018. Prenatal: window 1-4 (orange box), postnatal: window 6-9 (blue box). Cell types from Charles' 2022 study: 6 major inhibitory neuron cell types, 4 major excitatory principal neuron cell types, Microglia, Oligodendrocytes precursor cells (OPC), Mature Oligodendrocytes and Astrocyte. x-axis: development window W1-4 (fetal) and W6-9 (postnatal); y-axis: $\log_2(\text{RPKM}+1)$.

2.) The AI literature has seen lots of misleading (incorrectly interpreted) and non-replicable results, largely due to massive fishing expeditions that are not properly cross-validated or replicated on independent samples (this reviewer is an AI researcher). I would want to see evidence that the findings of this data mining analysis were quality controlled, checked, an ideally replicated (this is admittedly very difficult for a study of this magnitude, so I welcome thoughts of the authors on his issue). Although the tissue-based experiments would presumably alleviate this concern to some extent, they are very small, compared to the magnitude of data mining that was performed. At least some basic testing of the findings using interpretable metrics should be presented.

Reply: The reviewer raises an important point. We equally see validation and replication as key challenges in biomedical AI research. In the present study, we experienced the validation part as especially challenging since a partial gold standard exists, the NIH MICAD (which our algorithm was trained on), and also because this pipeline was designed to be as sensitive as possible at answering a specific question that was not the original goal of MICAD; the radiotracer-protein association. Nevertheless, the reviewer's request that our findings of these data mining analysis were quality controlled, checked, and replicated are well received. We see four synergistic actions, the use of the best possible methods, the control of the results, the introduction of quality metrics, and the future development of control algorithms:

1.) In our study we used thorough methods to aim for high quality results:

- we downloaded the maximum data available with the entire pubmed,
- we used the best-possible training set with the NIH MICAD, and
- we trained our model to classify abstract texts describing a radiotracer with an average precision of 97.8 %, an average recall of 98.8 %, and an average F1 score of 98.3%.

2.) With experts in molecular imaging and radiopharmacy, we quality controlled and checked our results in multiple steps:

- we accepted the associations between proteins and genes with the help of ad-hoc literature searches and the agreement of at least two investigators,
- we individually verified all protein-to-gene translations using online databases such as genecards and pubgene, and
- we chose all proposed tracers individually following a set of rules from a clinical and radiopharmaceutical perspective in order to guarantee its possible application.

3.) Based on the reviewer's comment, we aimed to develop metrics that can assess the accuracy of literature retrieval in pubmed, and that can assess the accuracy of the automated protein-to-gene translation.

First, to estimate the accuracy of literature retrieval in pubmed, we assessed the accuracy of MICAD listed radiotracers from pubmed. In brief, we compared the 8139 *Imageable Genome* radiotracers to all 588 MICAD radiotracers via fuzzy matching between the tracer names. For all matches with a levenshtein similarity score >0.4 we extracted the 3 top ranked matches between MICAD tracer name and IG radiotracer_1 column, and the 3 top ranked matches for the radiotracer_2 column (supplementary table 2). Consequently, for each MICAD tracer, there are maximum 6 matches. Then, we manually verified the results.

However, we were not able to produce reliable test results, in our view based on four reasons that potentiate each other:

- Not all entries in the MICAD database .xlsx file point to available abstracts on pubmed because of copyright issues or because they have been retracted since MICAD was created.

- We only considered for classification the entries of pubmed that had an abstract available since we classified the abstract texts only, this approach discards pubmed entries without abstracts.
- The subsequent filtering step was based on the co-occurrence of RADIOTRACER and PROTEIN entities detected by the NER algorithm in the title or the abstract of the articles that were selected by the TC. This algorithm, has a combined accuracy for all entities between 75% and 90%, which is state-of-the-art in the field, but it will not detect several tracers and proteins correctly.
- Since we only investigated these explicit RADIOTRACER and PROTEIN associations, all papers reporting radiotracers and not mentioning proteins explicitly in the abstract or the title were outside our scope. We never used the information of any full text papers. MICAD on the contrary was a thorough, classic systematic review approach using the full text of a selection of papers that was performed differently.

Secondly, to assess the accuracy of the automated protein-to-gene translation, we used random sampling to extract 7 pools of 20 articles (in total 140 articles) from supplementary table 2, manually confirmed the tracer-disease-protein-gene associations, and verified if any tracer-gene associations have been approved by further studies. Overall, we found 3 articles with wrong annotations for protein and gene associations. These 3 wrong associations occurred in one random sample of 20. All other samples are 100% correctly annotated. Thus, the percentage for the whole dataset estimation is 2%.

We corrected the annotation accordingly in the manuscript.

Overall, we are convinced that approaches for quality control will need to be completely automatized, to be able to compute the amount of data created by novel AI-based approaches.

4.) One future possible external validation approach for example, though possibly even more complex and resource demanding than the *Imageable Genome* project itself, could consist of a systematic search on the literature of associations between the diseases that the imageable genome links to each radiotracer and said radiotracers, to determine how many of the proposed new indications have already been studied and verified. The repetition of this process after the publication and public availability of the *Imageable Genome* database could also serve as a measure of its impact on the biomedical research community.

We added some changes on the discussion addressing this issue, clarifying the scope of our research pipeline, its possible validation, limitations and comparability concerns to any existing gold standard, such as the MICAD database as follows:

“Our pipeline identified 6’387 publications describing 9’285 radiotracer-to-gene associations.”

(Page 6)

and:

“Yet, such meta-analyses will increasingly rely on the validity of claims within the compiled literature and on the subsequent validation analyses of the outcome of these human-AI pipelines. New challenges such as replicability issues will raise as a consequence of the magnitude of these results and the lack of external validation datasets, and they will require the collaborative efforts of the scientific community and novel external proofing techniques.”

(Page 19)

3.) It is not clear what that this argument in the Introduction is valid: “One reason for this translational bottleneck is linked to the fundamental lack of knowledge concerning the entirety of molecules that can be targeted with the repertoire of available molecular imaging agents”. These radiotracers that don’t make it to the clinic have been developed and their use has been demonstrated---cost, availability, and other factors are likely to limit clinical adoption. In general, the paper tends to over-claim, at times (for example, “The Imageable Genome can serve as a ‘Rosetta stone’ that systematically translates these complex genomic discoveries into easily applicable clinical imaging tests” claims to address the miniscule rates of translation of genomic findings into FDA-regulated tests. However it is not clear whether and how this primarily data mining study will change this figure).

Reply: We appreciate the reviewer’s question, and aim to answer with a practical example: The radiotracer F18-fluorocholin was initially developed for prostate cancer imaging, but is currently in this function replaced by PSMA tracers, which are clinically superior. However, F18-fluorocholin is “accidentally” also an excellent agent to detect parathyroid adenoma, and is now having a “second career” in the work-up hyperparathyroidism.

We believe that among the thousands of already developed radiotracers there are many radiotracers that would qualify for a second career if only a new purpose can be assigned. We also believe that genomic data are the key to repurposing radiotracers, and we see the main strength of the *Imageable Genome* in systematically being able to translate genomic findings into new purposes for already existing tracers, which will hopefully lead to a higher rate of radiotracers translated into the clinic.

Being involved in radiotracer development, the authors’ experience is a rather low cross-talk between radiopharmacy and genomics. As the *Imageable Genome* can translate between these fields that speak the languages of chemistry and biology, respectively, we came up with the comparison to the Rosetta stone.

4.) Much of the value of the current study relies on the availability of an easily accessible and interpretable software or web implementation to disseminate the findings. This is not discussed in the paper.

Reply: We completely agree with the reviewer, and indeed aim to provide a web implementation to disseminate the findings. For this purpose, we registered the domain www.imageablegenome.com and intend to apply for funding in order to create and maintain a user-friendly website for clinicians and researchers. Additionally, we are uploading the complete code of the NLP pipeline and the genetic analyses to a github repository that we will keep updated and to which we will upload future versions of the code as well as new tools.

5.) The naming of the tracers should probably be checked carefully. I spotted one typo related to a tracer I am personally familiar with: it is [18F]NOS, and not [18F]NOS-9

Reply: We are grateful for the reviewer’s comment, and carefully checked all tracer names in the manuscript. To the best of our knowledge, the target of the mentioned radiotracer is “iNOS” or “NOS2” (Inducible Nitric oxide synthase), while the name of the radiotracer published in MICAD is “[18F]iNOS-9”. The reference paper named the radiotracer “[18F]9”, which could explain why MICAD named it “[18F]iNOS-9”.

Reviewer #3

The manuscript entitled, “The Imageable Genome,” details the process by which the authors used text classifiers to identify an updated set of PET-compatible imaging tracers/targets, including many studies, associations, and targets not in the current NIH-supported Molecular Imaging and Contrast Agent Database (MICAD). The authors go on to detail three case uses for The Imageable Genome dataset, specifically in the brain, heart, and cancer, and identify several novel uses for existing tracers. The work presented here is very exciting and is of high interest for its readily translatable clinical and basic science applications.

Reply: Thank you very much! We have been excited while working on the project during the last years, and we are now excited to present the completed project. We are especially delighted that the first readers of our manuscript share our excitement.

Below are some comments and questions related to the work presented in no particular order:

1. This work is very exciting, and it seems like it would be a shame to have the identifications buried in supplemental tables. It seems like this work could better serve the community as part of a standalone database (or an update to NIH MICAD) that could be updated as more is learned. The authors should strongly consider doing this.

Reply: We completely agree with the reviewer, and we are currently considering to serve the community in three ways:

We will contact the NIH MICAD and offer to help updating and expanding their database.

As we developed an AI that detects molecular imaging publications, we will also offer to help updating the Medline MESH term “molecular imaging”.

Finally, we aim to provide a web implementation to disseminate our findings. For this purpose, we registered the domain www.imageablegenome.com and intend to apply for funding in order to create and maintain a user-friendly website for clinicians and researchers. Additionally, we are uploading the complete code of the NLP pipeline and the genetic analyses to a github repository that we will keep updated and to which we will upload future versions of the code as well as new tools.

2. The authors chose to focus on neurology, cardiology, and oncology for the case uses as they identified many imageable genes already associated with these diseases (Fig. 2C, lines 113-121). As I understand it, the identification may be self-fulfilling as many of these tracers were developed for those purposes; however, the stated goal of “The Imageable Genome” is to expand these molecular imaging tools into new areas. Could the authors apply the same methodology to an organ and disorder not widely diagnosed using PET or PET-CT currently to demonstrate the untapped potential to community?

Reply: We picked up the challenge posed by the reviewer, and demonstrate how the *Imageable Genome* can be used to derive new imaging methods for COVID-19. Thereby, we used two recent studies:

Toni et al. (*NATURE* 2021) generated sc/snRNA-Seq atlases of SARS-CoV-2 infected lung (16 patients, 106,792 cells/nuclei, 24 specimens); heart (18 patients, 40,880 cells/nuclei, 19 specimens), liver (15 patients, 47,001 cells/nuclei, 16 specimens) and kidney (16 patients, 33,872 cells/nuclei, 16 specimens), in parallel to sc/snRNA-Seq data generated from 11 healthy lung donors and 21 healthy heart donors.

We mapped the imageable genes to the cell-type-specific transcriptional changes in lung or heart cell types associated with COVID-19 and show in **Extended Data Fig. 10a** the occupancy of imageable genes among the top 5 lung cell types reported to have the highest transcriptional alterations after SARS-CoV-2 infection; and in **Extended Data Fig. 10b** the 3 heart cell types mentioned in the cited paper (cardiomyocytes, pericytes or fibroblasts) plus 2 selected heart cell types from our side (macrophages and Vascular Endothelial cells) with significant transcriptional changes after SARS-CoV-2 infection.

Johannes et al. (*NATURE* 2021) generated a lung atlas that profiled 116,314 nuclei, including 79,636 nuclei from COVID-19-infected lungs (n=19) and 36,678 nuclei from control lungs (n=7). For four cell types, AT1, AT2, monocyte and macrophage, we cross the cell-type specific DEGs pooled from comparisons between COVID19-infected lungs versus healthy lungs to the DEGs pooled from Toni's paper and show the number of common DEGs between the two papers. From the common DEGs, we listed the Imageable common DEGs in **Extended Data Fig. 10c**.

Extended Data Fig. 10 The Imageable Genome in SARS-CoV-2 infected patients. **a,b**, Expression differences between COVID-19 and healthy lung and heart (Toni, Nature 2021) shown by volcano plots of significance ($-\log_{10}(\text{Pvalue})$) versus magnitude ($\log_2(\text{fold change})$) for each gene (dots) by cell type. *Imageable Genome* DEGs are highlighted in red and green color. Horizontal dashed line, blue, FDR < 0.05 ; black, FDR < 0.01 . **c**, Venn diagrams of the number of overlapping lung cell-type specific DEGs among the three datasets: Toni, Nature 2021, Johannes, Nature 2021 and the *Imageable Genome*. For the *Imageable Genome* DEGs present in both papers, expression change of each DEG is shown by barplot. X-axis: $\log_2(\text{Fold change})$ between COVID-19 lungs versus healthy lungs).

We included this part into the manuscript as follows:

“THE IMAGEABLE GENOME IN COVID-19

Diagnosing COVID-19, identifying affected tissues, and understanding its impact on human health remain a global health care challenge. Analysing 106’792, 40’880, 47’001, and 33’872 cells single-nucleus transcriptomes of SARS-CoV-2 infected lung, heart, liver and kidney, we identified 36 cell-type specific imageable genes up or down-regulated exclusively in COVID-19 versus healthy tissues (Extended Data Fig. 10). These results demonstrate how the Imageable Genome might allow expanding molecular imaging beyond neurology, cardiology, and oncology into new fields.

(Page 14)

3. On lines 169-170 (and it comes up again in the cardiology and oncology sections), the authors state, “Finally, while the Imageable Genome suggests promising imaging targets for already existing radiotracers, it also guides the development of novel radiotracers.” How exactly does this study guide the development of novel radiotracers? It seems to me that this is just a wish list of genes/proteins that it would be great to have a radiotracer that could be used to detect.

Reply: The reviewer is right. We meant to express that the *Imageable Genome* guides target selection for novel radiotracers. We corrected the wording accordingly on pages 9 and 11.

4. The authors often only identify how many differentially expressed genes are imageable. For example, as stated only 4% and 9% of differentially expressed genes in the embryo/adult heart stages are imageable. Would it be possible for the authors to compile a list of imageable genes (irrespective of differential expression) across different organs based on expression data from databases like GTEX, Human Protein Atlas, or similar?

Reply: Thanks for the comment. We apologize for the unclear demonstration in our **Figure 2a**, in which we actually have included the normalized expressions of Imageable genome genes across 24 healthy organs into the Circos plot, track 2. This data was retrieved from the ARCHS4 database (Henry E. Miller, BMC Bioinformatics 2021; Alexander Lachmann, Nat Commun 2018) and shown by a circle heatmap. In addition, we have presented the gene expression correlation analysis into track 4. The original paper (Henry E. Miller, BMC Bioinformatics 2021) has used GTEx database to train and validate their correlation analysis for GEO database.

We have prepared the healthy organ expression values into **Supplementary Table 3**, and updated the manuscript to make it clear for readers:

“The *Imageable Genome* is constantly growing, and currently comprises 1’166 genes located on all chromosomes but the Y chromosome (**Fig. 2a**), and 7 protein-coding genes on human mitochondrial DNA. It has a diverse expression pattern across different healthy organs⁸⁻¹⁰ (**Fig. 2a**, track 1, **Supplementary Table 3**), and high occupancy in major human diseases¹¹, with most imageable genes associated with multiple diseases (**Fig. 2a**, track 2).”

(Page 6)

FIGURE 2. The *Imageable Genome*. **a**, A circos plot depicting 1'166 out of 1'173 genes of the *Imageable Genome* with their chromosome locations (7 mitochondrial genes not shown): track 1, expression across 24 healthy tissues (red: relatively high gene expression, green: relatively low gene expression); track 2, gene-disease associations across 12 major diseases, with colour code corresponding to the disease classification in (c); track 3, number of radiotracers targeting an *Imageable Genome* gene. The height of red peak represents the corresponding number; track 4, *Imageable Genome* genes targeted by more than 30 radiotracers are labelled, and the links to their genome-wide co-expressed genes across 24 healthy tissues are highlighted in the innermost layer (pink, blue, purple and orange lines). **b**, Scatter plot summarizing a list of enriched gene ontology (GO) terms from 1'165 *Imageable Genome* genes (8 genes are not present in DAVID database). Rich factor is the ratio of the *Imageable Genome* gene number in a GO term to the total gene number. GO terms passing the thresholds $-\log_{10}(\text{FDR}) > 30$ and Rich factor $> 15\%$ are labelled. **c**, Disease classification of 916 imageable genes. Top enriched 12 major diseases and their subcategorized diseases are shown in a sunburst plot, with the area corresponding to the ratio of gene number within a major disease to the total gene number. 24 major healthy tissues used for circos plot track 1 from outside to inside: cartilage, esophagus, male reproductive tissue, liver, bone, kidney, cardiac, thyroid, muscle, stomach, respiratory, endothelial, intestines, adipose, prostate, prenatal, pancreas, skin, stem-like tissue, brain, mammary, immune, retina, female reproductive tissue.

5. Minor critique, I noticed that the text does not refer to all the panels of Figure 4 and that some of the panels are misattributed. Specifically, “we identified 6 imageable genes that are predictive for the onset of atrial fibrillation (Fig. 4d)”; however, this appears to Fig. 4E in the figure.

Reply: Thank you for the careful read. We fixed this point.

REVIEWERS' COMMENTS

Reviewer #1 (Remarks to the Author):

All my concerns have been addressed and I appreciate the extra work the authors have done.

Reviewer #2 (Remarks to the Author):

The authors have made extensive revisions and/or provided convincing responses to all comments. In response to the small sample size concerns, they have now provided additional data comparing their findings of imageable genome to analogous recent studies, showing relatively good agreement. The authors did register a new domain in order to host a web site, but no site is currently accessible, so I cannot evaluate the ultimate accessibility of these pipelines and results to the public, until such web site is actually implemented.

Reviewer #3 (Remarks to the Author):

The authors have done a tremendous job addressing the comments and concerns of the other reviewers and myself. I have no further comment.

REVIEWERS' COMMENTS

Reviewer #1

1.) All my concerns have been addressed and I appreciate the extra work the authors have done.

Reply: Thank you very much. We equally appreciate the work of the reviewer.

Reviewer #2

1.) The authors have made extensive revisions and/or provided convincing responses to all comments.

Reply: We are grateful for all of these comments, as they indeed helped to improve our manuscript.

2.) In response to the small sample size concerns, they have now provided additional data comparing their findings of imageable genome to analogous recent studies, showing relatively good agreement.

Reply: Thank you again for the challenge. We are happy it worked out well.

3.) The authors did register a new domain in order to host a web site, but no site is currently accessible, so I cannot evaluate the ultimate accessibility of these pipelines and results to the public, until such web site is actually implemented.

Reply: We handed over all our methods, all raw data, and all results to the journal, and made our code freely available on GitHub. We feel that we need to develop new original content first, before setting up the web site.

Reviewer #3

1.) The authors have done a tremendous job addressing the comments and concerns of the other reviewers and myself. I have no further comment.

Reply: Thank you very much for your interest in our research, and your help in improving it!